

# Global distribution of methane emissions, emission trends, and OH concentrations and trends inferred from an inversion of GOSAT satellite data for 2010-2015

Joannes D. Maasakkers[1], Daniel J. Jacob[1], Melissa P. Sulprizio[1], Tia R. Scarpelli[1], Hannah Nesser[1], Jian-Xiong Sheng[1], Yuzhong Zhang[1], Monica Hersher[1], A. Anthony Bloom[2], Kevin W. Bowman[2], John R. Worden[2], Greet Janssens-Maenhout[3], and Robert J. Parker[4,5,6]

[1]Harvard University, Cambridge, Massachusetts 02138, United States
[2]Jet Propulsion Laboratory, California Institute of Technology, Pasadena, CA, USA
[3]European Commission Joint Research Centre, Ispra (Va), Italy
[4]Earth Observation Science, Department of Physics and Astronomy, University of Leicester, Leicester, UK
[5]Leicester Institute for Space and Earth Observation, University of Leicester, Leicester, UK
[6]NERC National Centre for Earth Observation, UK

**Correspondence:** J.D. Maasakkers (maasakkers@fas.harvard.edu)

**Abstract.** We use 2010-2015 observations of atmospheric methane columns from the GOSAT satellite instrument in a global inverse analysis to improve estimates of methane emissions and their trends over the period, as well as the global concentration of tropospheric OH (the hydroxyl radical, methane's main sink) and its trend. Our inversion solves the Bayesian optimization problem analytically including closed-form characterization of errors. This allows us to (1) quantify the information content

from the inversion towards optimizing methane emissions and its trends, (2) diagnose error correlations between constraints on emissions and OH concentrations, and (3) generate a large ensemble of solutions testing different assumptions in the inversion. We show how the analytical approach can be used even when prior error standard deviation distributions are log-normal. Inversion results show large overestimates of Chinese coal emissions and Middle East oil/gas emissions in the EDGAR v4.3.2 inventory, but little error in the US where we use a new gridded version of the EPA national greenhouse gas inventory as

prior estimate. Oil/gas emissions in the EDGAR v4.3.2 inventory show large differences with national totals reported to the United Nations Framework Convention on Climate Change (UNFCCC) and our inversion is generally more consistent with the UNFCCC data. The observed 2010-2015 growth in atmospheric methane is attributed mostly to an increase in emissions from India, China, and areas with large tropical wetlands. The contribution from OH trends is small in comparison. We find that the inversion provides strong independent constraints on global methane emissions (546 Tg a$^{-1}$) and global mean OH

concentrations (atmospheric methane lifetime against oxidation by tropospheric OH of $10.8 \pm 0.4$ years), indicating that satellite observations of atmospheric methane could provide a proxy for OH concentrations in the future.



# 1 Introduction

Methane is an important greenhouse gas with a particularly strong decadal climate impact (Stocker et al., 2013). The atmospheric methane concentration has increased by a factor of 2.5 since pre-industrial times (Hartmann et al., 2013). This increase is not well understood but must be mainly driven by anthropogenic sources including the oil/gas industry, coal mining, live-
stock, landfills, wastewater treatment, and rice cultivation (Dlugokencky et al., 2011; Kirschke et al., 2013; Saunois et al., 2016). Wetlands are the main natural source and could be affected by climate change (Kirschke et al., 2013). Atmospheric methane has a lifetime of $9.1 \pm 0.9$ years (Prather et al., 2012), with a dominant sink from oxidation by the hydroxyl radical (OH) that is also subject to interannual variability and trends (Holmes et al., 2013). Methane concentrations rose by $\sim 1$ % $a^{-1}$ in the 1980s and early 1990s, plateaued in the early 2000s, and have resumed increasing at $\sim 0.4$ % $a^{-1}$ since 2007
(esrl.noaa.gov/gmd/ccgg/trends_ch4), for reasons that remain unclear (Turner et al., 2017). Inverse analyses can help interpret these trends by combining atmospheric methane observations with a chemical transport model (CTM) to infer the distribution of methane emissions most likely to explain the observations (Houweling et al., 2017; Saunois et al., 2016; Jacob et al., 2016). Here we use global 2010-2015 methane observations from the GOSAT satellite in an analytical inverse analysis with full error characterization to better quantify methane sources and interpret the recent trend, including changes in both methane emissions
and OH concentrations.

A number of explanations have been proposed for the renewed growth of atmospheric methane concentrations since 2007. A parellel increase in ethane has been proposed as evidence for an increase in oil/gas emissions (Hausmann et al., 2016; Franco et al., 2016). A trend towards isotopically lighter methane has been attributed to an increase in microbial sources such as
livestock and wetlands (Schaefer et al., 2016; Schwietzke et al., 2016; Nisbet et al., 2016; McNorton et al., 2016). Worden et al. (2017) suggest that a decrease in open fire emissions may mask the isotopic signature of increasing fossil fuel emissions. Observations of methyl chloroform, a proxy for global OH concentrations, suggest that a decrease in the methane sink may be implicated in the renewed growth (Turner et al., 2017; Rigby et al., 2017; McNorton et al., 2018). Turner et al. (2017) find that the surface record of methane observations is too sparse to arbitrate between these different explanations. GOSAT satellite
observations used here provide much denser global coverage.

GOSAT was launched in 2009 and measures atmospheric methane columns with high precision (0.7 %) by solar backscatter in the shortwave infrared (SWIR) (Butz et al., 2011; Buchwitz et al., 2015; Kuze et al., 2016). A number of inverse analyses have used the GOSAT data to improve estimates of methane emissions (Monteil et al., 2013; Cressot et al., 2014; Alexe et al.,
2015; Turner et al., 2015; Pandey et al., 2016, 2017). Here we use the GOSAT data to optimize not only emissions but also their 2010-2015 trends together with OH concentrations and their trends. The independent optimization of OH and emissions in the inversion is based on the different signatures of those two terms on the methane concentration fields (Zhang et al., 2018). We use an analytical inverse method with closed-form error characterization of the solution, rather than the adjoint approaches used in previous inverse studies that do not provide rigorous characterization of errors. This allows us in particular to diagnose



the error correlation between the independent constraints on methane emissions and OH concentrations and their trends. It also allows us to readily conduct inversions for an ensemble of cases once the Jacobian matrix for the problem has been constructed.

## 2   Data and methods

We use the GEOS-Chem CTM (www.geos-chem.org) as forward model to simulate the distribution of atmospheric methane and its response to trends. Model results are fit statistically to the GOSAT data by Bayesian optimization, including regularization from prior knowledge of methane emissions and OH concentrations. The January 2010 - December 2015 GOSAT methane column data are arranged in an observation vector $\mathbf{y}$, and the inversion optimizes a state vector $\mathbf{x}$ including global methane emissions on the $4° \times 5°$ GEOS-Chem grid, 2010-2015 linear trends of emissions on that same grid, and global mean OH concentrations for individual years (we will also present results from an inversion optimizing a linear OH trend over the 2010-2015 period). The optimal solution $\hat{\mathbf{x}}$ is obtained by minimizing a Bayesian cost function that balances the information from the observations (weighed by the observational error covariance matrix $\mathbf{S_O}$) and the prior knowledge $\mathbf{x_a}$ (weighed by the prior error covariance matrix $\mathbf{S_a}$) (Rodgers, 2000). Below we describe the different elements and steps in the inversion.

### 2.1   GOSAT observations

The TANSO-FTS instrument onboard the Greenhouse Gases Observing Satellite (GOSAT) observes column-averaged dry-air methane mixing ratios by solar back-scatter in the SWIR with near-unit sensitivity down to the surface (Butz et al., 2011). The satellite is in polar sun-synchronous orbit. Observations are made at around 13:00 local time for circular pixels of 10 km diameter. In the default observation mode, the pixels are separated by ∼250 km along-track and cross-track, with repeated observation of the same pixels every 3 days. Denser observations are also made in target mode over features of interest. GOSAT observations have shown no significant drift or degradation of data quality since the beginning of the record (Kuze et al., 2016). We use the University of Leicester version 7 $CO_2$ proxy retrieval over land (Parker et al., 2011, 2015) from January 2010 to December 2015 in order to have even observations of all seasons. The single-observation precision is 13 ppb and the relative (regional) bias is 2 ppb compared to ground-based column-averaged dry-air mole factions from the Total Carbon Column Observing Network (TCCON) (Buchwitz et al., 2015). Figure 1 illustrates the GOSAT data ingested in our inversion, representing a total of 1,211,468 retrievals. Glint data over the oceans and data poleward of 60° are not included because of seasonal sampling biases (Turner et al., 2015).

### 2.2   Prior estimates

The inversion requires prior estimates and error statistics for all components of the state vector including methane emissions on the $4° \times 5°$ GEOS-Chem grid (1009 ice-free land-containing grid cells with prior emissions larger than $8 \times 10^{-3}$ Mg km$^{-2}$a$^{-1}$, covering 99% of global emissions), 2010-2015 linear emission trends on the same grid, and global mean OH concentrations



for individual years 2009-2015 (2009 is only used for initialization), for a total of 2025 state vector elements.

Table 1 gives our global prior inventory with the contributions from different source types, and Figure 2 shows the spatial distributions. Monthly wetlands emissions for individual years are from the WetCHARTS v1.0 extended ensemble mean (Bloom et al., 2017). For anthropogenic emissions we use the EDGAR v4.3.2 global emission inventory for 2012 (edgar.jrc. ec.europa.eu) as worldwide default), including additional information from EDGAR to subset the 'fuel exploitation' emissions category into oil/gas and coal mining. Over the continental US, we replace EDGAR v4.3.2 with a gridded version of the US EPA greenhouse gas inventory(Maasakkers et al., 2016). In Canada and Mexico, we use the oil/gas emissions from Sheng et al. (2017). Anthropogenic emissions are aseasonal except for manure management and rice cultivation. Seasonal scaling of manure management emissions is done using the temperature dependence of Maasakkers et al. (2016). Seasonal scaling of rice cultivation emissions is based on Zhang et al. (2016). Daily global open fire emissions are from QFED (Darmenov and da Silva, 2013). Termite emissions are from Fung et al. (1991). Emissions from geological macroseeps (oil/gas seeps and mud volcanoes) are based on Etiope (2015) and Kvenvolden and Rogers (2005). For areal seepage, we use the sedimentary basins (microseepage) and potential geothermal seepage maps from Kvenvolden and Rogers (2005) with the emission factor previously used by Lyon et al. (2015). Over the US, we use the sedimentary basins map from the Energy Information Administration (2016) and basin-specific emission factors from Etiope and Klusman (2010).

Construction of the prior error covariance matrix $S_a$ requires estimates of error variances for the prior emissions on the $4° \times 5°$ grid. For wetland emissions, we use the standard deviation of $4° \times 5°$ annual emissions from the WetCHARTs ensemble members (Bloom et al., 2017). The error variance averages 58% on the grid-level. For US anthropogenic emissions and oil/gas emissions in Canada and Mexico, we use the scale-dependent error variances from Maasakkers et al. (2016). For lack of better information, we assume 50% error standard deviation for EDGAR v4.3.2 emissions (Turner et al., 2015) and 100% for non-wetland natural emissions. The diagonal terms of $S_a$ are then constructed by adding the error variances of individual source types for $4° \times 5°$ grid cells in quadrature, capping total errors at 50%. We assume no error spatial covariance on the $4° \times 5°$ grid so that $S_a$ is diagonal. This is a reasonable assumption for anthropogenic emissions (Maasakkers et al., 2016), though errors on wetland emissions may still be correlated on that scale (Bloom et al., 2017).

Our state vector in the inversion includes linear emission trends for $4° \times 5°$ grid cells over the 2010-2015 period, superimposed on interannual variability in the case of wetlands and fires. Our global prior estimate of mean methane emissions for the 2010-2015 period exceeds the sinks by 2.4% (Table 1), which would drive a 0.3% $a^{-1}$ increase in methane concentrations over that period even in the absence of an emission trend. Therefore our prior estimate of linear emission trends for individual $4° \times 5°$ grid cells is zero, with an absolute error standard deviation of 10% of the local prior emissions over the 2010-2015 time period (1.7% $a^{-1}$). This error standard deviation is based on trend estimates for North America inferred from GOSAT data (Turner et al., 2016; Sheng et al., 2018a).



**Table 1.** Prior global estimates of methane sources and sinks (mean 2010-2015 values).

| Source (Tg a$^{-1}$) | | Sink (Tg a$^{-1}$) | |
|---|---|---|---|
| **Natural** | | | |
| Wetlands | 161 | Tropospheric OH | 475 |
| Open fires | 15 | Stratospheric loss | 33 |
| Termites | 12 | Soil uptake | 18 |
| Seeps | 5 | Tropospheric Cl | 9 |
| | | | |
| **Anthropogenic** | | | |
| Livestock | 117 | | |
| Oil and Natural Gas | 70 | | |
| Coal Mining | 38 | | |
| Rice Cultivation | 38 | | |
| Wastewater | 38 | | |
| Landfills | 30 | | |
| Other Anthropogenic[a] | 25 | | |
| | | | |
| Total Source | 548 | Total Sink | 535 |

[a] including fossil fuel combustion, industrial processes, and agricultural field burning

The prior estimate of the global tropospheric OH concentration is based on a GEOS-Chem full-chemistry simulation (Wecht et al., 2014) that yields a methane lifetime $\tau_{CH_4}^{OH}$ of 10.6 years, consistent with the best estimate inferred from the methyl chloroform proxy (Prather et al., 2012). Here and elsewhere, $\tau_{CH_4}^{OH}$ is defined as the ratio between the total mass of atmospheric methane (including the stratosphere) and the annual loss rate from oxidation by OH below the tropopause. The uncertainty in the methane lifetime is about 10% (Prather et al., 2012) but the uncertainty on OH interannual variability is less, about 3% (Holmes et al., 2013). We assume a 3% error standard deviation in the global annual mean OH concentration for our standard inversion but also conduct a sensitivity study with 10% error standard deviation. We further conduct an inversion taking the OH trend over the 2010-2015 period as linear, and assuming in that case error standard deviations of 10% for the mean global OH concentration and 5% a$^{-1}$ (absolute) for the linear trend. Scaling of global OH concentrations in the inversion is done without modifying the spatial or seasonal OH distribution. Zhang et al. (2018) found that inversions of atmospheric methane data using the 3-D GEOS-Chem OH fields give consistent results with inversions using other global OH distributions from the ACCMIP model ensemble (Naik et al., 2013).



## 2.3 Forward model

We use the GEOS-Chem CTM v11-01 at $4° \times 5°$ grid resolution (Wecht et al., 2014; Turner et al., 2015) as forward model for the inversion. The model is driven with 2009-2015 MERRA-2 meteorological fields(Bosilovich et al., 2016) from the NASA Global Modeling and Assimilation Office (GMAO). Atmospheric methane concentrations are initialized on January 2009 us-
5 ing the previous GOSAT inversion results of Turner et al. (2015), shown in that work to be unbiased compared to surface and aircraft background data including for the tropospheric meridional gradient.

The loss from oxidation by tropospheric OH is computed with archived 3-D monthly fields of OH concentrations from a GEOS-Chem full-chemistry simulation as described by Wecht et al. (2014). Local tropopause information is from the MERRA-
10 2 data. The global loss rate for individual years is optimized in the inversion by uniform scaling of the OH concentrations. Other minor loss terms include stratospheric oxidation computed with archived monthly loss frequencies from the NASA Global Modeling Initiative model (Murray et al., 2012), tropospheric oxidation by Cl atoms computed using archived Cl concentration fields from Sherwen et al. (2016) and the reaction rate constant from Allan et al. (2007), and soil uptake as described by Fung et al. (1991) with temperature-based seasonality based on Ridgwell et al. (1999). These minor sinks are not optimized in the
15 inversion.

The GEOS-Chem simulation of GOSAT methane columns features a latitude-dependent background bias that needs to be corrected (Turner et al., 2015). This bias likely reflects a model overestimate of methane in the extratropical stratosphere (Saad et al., 2016), and is common across global models due to excessive meridional transport in the stratosphere (Patra et al., 2011).
Stanevich (2018) found significant difference in methane columns simulated by GEOS-Chem at $4° \times 5°$ compared to $2° \times 2.5°$ resolution, but we find that this difference is mainly in the stratosphere (Appendix A). We remove the background bias by applying the latitudinal correction based on background grid cells from Turner et al. (2015), recomputed with the University of Leicester v7 GOSAT proxy retrieval (Parker et al., 2015) and the MERRA-2 meteorological fields. The mean model - GOSAT difference in column mean mixing ratio for background $4° \times 5°$ grid cells is fitted to a second order polynomial of latitude:

$$25 \quad \xi = \left(4.0\theta^2 - 1.3\theta\right) \times 10^{-3} - 5 \tag{1}$$

where $\theta$ is the latitude in degrees and $\xi$ is the model correction in ppb. This correction is similar to Turner et al. (2015) who used $\xi = \left(5\theta^2 - 5\theta\right) \times 10^{-3} - 0.5$. A seasonal bias remains after application of this correction and we fix it by removing the zonal monthly mean concentration differences averaged over rolling $12°$ latitudinal bands. This seasonal bias may be due to emissions or model errors in stratospheric or tropospheric transport (Saad et al., 2016; Bader et al., 2016; Stanevich, 2018). We
find that the seasonal correction does not affect the inversion results significantly, as shown in Appendix B where we optimize emissions for individual seasons separately without applying a seasonal correction.



## 2.4 Observational error covariance matrix

The observational error covariance matrix $\mathbf{S_O}$ includes contributions from random instrument and forward model errors. We construct it by the residual error method of Heald et al. (2004) using the 2010-2015 time series of local methane column differences $\Delta = y_{GEOS-CHEM,\,prior} - y_{GOSAT}$ for individual $4° \times 5°$ grid cells between the GEOS-Chem model with prior estimates (emissions and OH concentrations) and the GOSAT observations after background bias correction. The mean difference $\overline{\Delta} = \overline{y_{GEOS-CHEM,\,prior} - y_{GOSAT}}$ is to be corrected in the inversion while the residual error $\Delta' = \Delta - \overline{\Delta}$ is taken as the observational error. Statistics of $\Delta'$ define the observational error variance (diagonal of the observational error covariance matrix). The same method was previously used in the satellite-based methane inversions by Wecht et al. (2014) and Turner et al. (2015). The resulting observational error standard deviation averages 13 ppb. The mean instrument error standard deviation is 11 ppb (Parker et al., 2015), implying that most of the observational error is generally from the instrument rather than from the forward model. This would indeed be expected for the random error of individual measurements. For a given measurement, if the local error standard deviation computed by the residual error method is smaller than the reported measurement precision, then we use the latter instead; this is the case for 10% of retrievals. All observational error standard deviations are set to be at least 10 ppb (this threshold affects 8% of retrievals). $\mathbf{S_O}$ is taken to be diagonal for lack of better information but the general effect of error correlation in the observations is accounted for in the inversion by a regularization factor (Section 2.5).

## 2.5 Inversion procedure

We perform inversions with two different specifications of prior error variance statistics: normal and log-normal. Assumption of normally distributed errors enables a linear optimization problem with an analytical solution including full error characterization (Rodgers, 2000). Assumption of log-normal errors may be more appropriate for modeling the high tail of the probability density function (Zavala-Araiza et al., 2015) and also has the advantage of enforcing positive solutions (Miller et al., 2014), but the optimization problem is then non-linear. By comparing the two approaches we can evaluate consistency in results.

Both inversions minimize the Bayesian cost function $J(\mathbf{x})$ (Rodgers, 2000):

$$J(\mathbf{x}) = (\mathbf{x} - \mathbf{x_a})^T \mathbf{S_a}^{-1} (\mathbf{x} - \mathbf{x_a}) + \gamma (\mathbf{y} - \mathbf{F}(\mathbf{x}))^T \mathbf{S_O}^{-1} (\mathbf{y} - \mathbf{F}(\mathbf{x})) \tag{2}$$

where $\mathbf{x}$ is the state vector, $\mathbf{x_a}$ is the prior estimate, $\mathbf{S_a}$ is the prior error covariance matrix, $\mathbf{F}(\mathbf{x})$ is the simulation of observations $\mathbf{y}$ by the GEOS-Chem model, $\mathbf{S_O}$ is the observational error covariance matrix, and $\gamma$ is a regularization factor (Brasseur and Jacob, 2017). Zhang et al. (2018) showed in an observing system simulation experiment (OSSE) for inversion of methane satellite data that a regularization factor $\gamma = 0.05$ was needed to prevent overfitting because of correlation in the observational error that is missing from the diagonal formulation of $\mathbf{S_O}$ and is otherwise difficult to quantify. Diagnosis of overfit and optimization of $\gamma$ is readily done in an OSSE such as in Zhang et al. (2018) where the "true" solution is known.





Here we find that using $\gamma = 1$ (as in the pure Bayesian statement of the optimization problem) produces checkerboard patterns in the solution that are likely spurious. We choose $\gamma = 0.05$ for our base inversion as providing the best balance between prior and observational terms in the posterior value of the cost function, and examine the sensitivity to the choice of $\gamma$ by conducting a sensitivity inversion with $\gamma = 0.1$.

Further balancing of the cost function is needed because the global OH concentration and its interannual variability are represented by only 7 state vector elements, while the emissions on the $4° \times 5°$ grid are represented by 1009 elements. To provide equal weight to OH and emissions for explaining global methane trends, we increase the weight of the OH terms in the cost function (through the OH components of $\mathbf{S_a}$) by the ratio of the number of state vector elements $1009/7$. The sensitivity inversion assuming 10% prior error standard deviation on OH instead of 3% is equivalent to decreasing this weighting by a factor of 11.

The GEOS-Chem forward model $\mathbf{y} = \mathbf{F}(\mathbf{x})$ relating methane column concentrations $\mathbf{y}$ to the state vector $\mathbf{x}$ is essentially linear. There is a small non-linearity from the optimization of OH concentrations because changes in the methane concentrations affect the loss rate (Houweling et al., 2017) which we neglect because changes in methane concentrations are small. We therefore express the forward model as $\mathbf{F}(\mathbf{x}) = \mathbf{Kx} + \mathbf{c}$ where $\mathbf{K} = \partial y/\partial x$ is the Jacobian matrix of the model and $\mathbf{c}$ is an initialization constant (January 2009 concentrations taken from Turner et al. (2015)). Replacing $\mathbf{F}(\mathbf{x}) = \mathbf{Kx}$ in Equation 2 and subtracting the initialization constant $\mathbf{c}$ from the observations, the minimization problem $dJ(\mathbf{x})/d\mathbf{x} = \mathbf{0}$ has an analytical solution for the optimal posterior solution $\widehat{\mathbf{x}}$ (Rodgers, 2000):

$$20 \quad \widehat{\mathbf{x}} = \mathbf{x_a} + \mathbf{S_a}\mathbf{K}^T \left( \mathbf{KS_aK}^T + \frac{\mathbf{S_O}}{\gamma} \right)^{-1} (\mathbf{y} - \mathbf{Kx_a}) \quad (3)$$

The posterior error covariance matrix $\widehat{\mathbf{S}}$ describing the error statistics of $\widehat{\mathbf{x}}$ is given by:

$$\widehat{\mathbf{S}} = \left( \gamma \mathbf{K}^T \mathbf{S_O}^{-1} \mathbf{K} + \mathbf{S_a}^{-1} \right)^{-1} \quad (4)$$

and the averaging kernel matrix ($\mathbf{A} = \partial\widehat{\mathbf{x}}/\partial\mathbf{x}$) defining the sensitivity of the solution to the true state is given by:

$$\mathbf{A} = \mathbf{S_a}\mathbf{K}^T \left( \mathbf{KS_aK}^T + \frac{\mathbf{S_O}}{\gamma} \right)^{-1} \quad (5)$$

The trace of the averaging kernel matrix defines the degrees of freedom for signal (DOFS) of the inversion, that is the number of pieces of information on the state vector that can be gained from the observing system.

The analytical solution as described by Equations 3-5 requires the explicit construction of the Jacobian matrix $\mathbf{K}$ characterizing the GEOS-Chem model. We do this column-by-column with GEOS-Chem simulations perturbing independently each element of the state vector. This is readily achievable even for 2025 state vector elements as a massively parallel computation. Sparse matrix algebra is used where possible in solving Equations 3-5, taking advantage of the diagonal structure of the error



covariance matrices.

The analytical solution to the Bayesian optimization problem requires assumption of Gaussian errors, but this allows for the possibility of negative values of state vector elements. Small negative emissions could conceivably be attributed to soil uptake, but large negative emissions are most likely unphysical (Miller et al., 2014). We can address this problem in the Bayesian solution by optimizing for $\ln(\mathbf{x})$ instead of $\mathbf{x}$, with normal Gaussian errors specified for $\ln(\mathbf{x})$ (corresponding to log-normal errors for $\mathbf{x}$). The model is then non-linear, so that the solution and the corresponding error statistics must be found iteratively with an updated Jacobian matrix $\mathbf{K}'_N = \partial \mathbf{y}/\partial \ln \mathbf{x}$ at each iteration $N$. This recomputation is immediate using the previously derived Jacobian matrix $\mathbf{K}$ for the linear problem, since the individual scalar elements $\partial y_i/\partial \ln(x_i)$ of $\mathbf{K}'$ are related to those of $\mathbf{K}$ by $\partial y_i/\partial \ln(x_j) = x_j \partial y_i/\partial x_j$. Thus only a simple scaling of the linear Jacobian matrix is required at each iteration. This conversion to log space is done only for the emissions component of $\mathbf{x}$. Emission trends and global OH concentrations are still optimized with normal error distributions and no scaling is applied to those rows of the Jacobian.

The iterative solution for the inverse problem with lognormal errors is obtained with the Levenberg-Marquardt method (Rodgers, 2000) for each iteration $N$:

$$\mathbf{x}'_{N+1} = \mathbf{x}'_N + \left((1+\kappa)\mathbf{S'_A}^{-1} + \gamma \mathbf{K}_N'^{T}\mathbf{S_O}^{-1}\mathbf{K}_N'^{-1}\right)^{-1}\left(\gamma\mathbf{K}_N'^{T}\mathbf{S_O}^{-1}(\mathbf{y}-\mathbf{K}\mathbf{x}_N) - \mathbf{S'_A}^{-1}(\mathbf{x}'_N - \mathbf{x'_A})\right) \tag{6}$$

where $\mathbf{x}' = \ln\mathbf{x}$, the initial guess $\mathbf{x}'_0$ is the prior estimate, and $\kappa$ is a coefficient for iterative approach to the solution that is set to 100 to start and is gradually decreased as the solution is approached. The prior error covariance matrix $\mathbf{S_a}'$ (diagonal elements $s'_A$) defining error variances for $\ln\mathbf{x_a}$ is derived from the perviously described prior error covariance matrix $\mathbf{S_a}$ (diagonal elements $s_A$) by scaling the error variances for the individual elements:

$$s'_A = \left(\frac{\ln\left(\frac{x_A+\sqrt{s_A}}{x_A}\right) + \left|\ln\left(\frac{x_A-\sqrt{s_A}}{x_A}\right)\right|}{2}\right)^2 \tag{7}$$

## 2.6 Error correlations between global estimates of sources and sinks

Inversion results for the spatial distributions of emissions and trends on the $4° \times 5°$ grid are mainly informed by local/regional patterns of methane concentration. However, implied inversion results for the global methane emission and its trend may be significantly correlated with those for the global tropospheric OH concentration and its trend. Some separation is expected because sources of methane have a different imprint on the global methane distribution than the OH sink (Zhang et al., 2018) but it is important to quantify the error correlation, i.e., the extent to which adjustments to the global methane emission and its trend may be aliased by adjustments to the global OH concentration and its trend.





To do this we reduce the dimensionality of the inverse analysis by collapsing global emissions and trends into one state vector element each. Following Calisesi et al. (2005), if the state vector can be transformed using a summation matrix $\mathbf{W}$ as:

$$\mathbf{x_{red}} = \mathbf{W}\mathbf{x} \tag{8}$$

then the averaging kernel matrix of the reduced system ($\mathbf{A_{red}}$) is given by:

$$\mathbf{A_{red}} = \mathbf{W}\mathbf{A}\mathbf{W}^* \tag{9}$$

where $\mathbf{W}^*$ is the generalized pseudo-inverse of $\mathbf{W}$ $\left((\mathbf{W}^T\mathbf{W})^{-1}\mathbf{W}^T\right)$. Our original state vector $\mathbf{x}$ in this case includes mean 2010-2015 emissions and their linear trends on the $4° \times 5°$ grid, and the global mean tropospheric OH concentration for 2010-2015 and its linear trend. Again, the minor sinks in Table 1 are not optimized and are maintained instead at their prior values. We apply the summation matrix $\mathbf{W}$ to the emission terms and thus reduce the state vector to four elements defining

the global methane budget (global mean emission, global mean OH concentration, global emission trend, global OH trend). The off-diagonal terms of the reduced averaging kernel matrix $\mathbf{A_{red}}$ then measure the extent to which differences relative to the true state are aliased between sources and sinks in the optimization of this global budget. The advantage of this summation approach, as compared to a global inversion including just four elements, is that the distribution of methane emissions and its trends is still optimized.

## 3   Results and Discussion

We conduct an ensemble of inversions to characterize the sensitivity of the solution to different assumptions made in the formulation of the inverse problem. Our base inversion optimizes annual mean emissions with normal error distributions and seasonal background correction to the GOSAT-model difference as discussed above. To test whether choices in the regulariza-

tion and cost function construction affect our conclusions we also conduct inversions with (1) log-normal error distributions for emissions, (2) a regularization factor $\gamma$ of 0.1 instead of 0.05, (3) no seasonal background correction to the model-GOSAT difference, (4) a 10% error standard deviation on the global OH concentration instead of 3%, (5) optimizing a linear trend in global OH concentration instead of year-to-year variability, assuming 10% error standard deviation for mean OH and 5% for the 2010-2015 trend, (6) no interannual variability in prior emission estimates, and (7,8) seasonally-resolved emission

optimization including seasonal correction and not (see Appendix B). All inversions produce consistent results and we will focus our main presentation on the base inversion, bringing in the sensitivity inversions to illustrate the spread of results and to address specific issues.

Before presenting results from the inversion, we compare the posterior solution to observations to show that the inversion

accomplishes its task of providing an improved forward model fit to observations. Figure 3 (top and middle panels) shows the improvement in the GEOS-Chem comparison to the GOSAT data when using posterior vs. prior emissions, emission trends,



and OH concentrations. As expected for a successful inversion, the posterior values provide a better fit to the observations. The inversion corrects prior underestimates over tropical regions and an overestimate over China. It also fits the observed 2010-2015 trend in methane concentrations and its latitudinal distribution. It does not fully correct the prior bias in the Arctic because GOSAT observations north of 60°N are not used in the inversion.

Figure 3 also shows independent evaluation of the inversion results with background observations from the NOAA co-operative flask sampling network (esrl.noaa.gov/gmd/ccgg/flask.php), the HIPPO aircraft campaigns across the Pacific and Atlantic (legs III-V, hippo.ornl.gov(Wofsy, 2011)), and the Total Carbon Column Observing Network network (TCCON, tccondata.org(Wunch et al., 2011)). These observations are mainly of the seasonal/latitudinal methane background and are not used in the inversion. The background is already well simulated in the prior estimate, and the posterior simulation does not degrade this agreement.

## 3.1 Spatial distribution and source attribution of methane emissions

Figure 4 shows the global distribution of mean 2010-2015 posterior emissions from the base inversion and from the sensitivity inversion assuming log-normal errors in the prior emission estimates. Correction patterns are very similar between the two inversions. Small negative emissions are found in the base inversion for 6 of the 1009 optimized grid cells. The inversion assuming log-normal errors does not allow these negative emissions. Downward corrections tend to be smaller in the inversion assuming log-normal errors, while positive corrections are larger and more concentrated in a few grid cells, as would be expected from the different shapes of the error standard deviation distributions.

The top right panel of Figure 4 shows the diagonal terms of the averaging kernel matrix for the base inversion (averaging kernel sensitivities), measuring the ability of the observations to constrain the inversion. The trace of the averaging kernel matrix (DOFS = 128) measures the number of independent pieces of information constrained by the inversion. A Bayesian inversion without correcting for overfit ($\gamma = 1$ in Equation 3) would erroneously produce much higher DOFS. We find that the inversion provides strong constraints on the $4° \times 5°$ grid for source regions in East Asia, central Africa, and South America. Averaging kernel sensitivities are generally weaker over North America and in Europe, indicating that the inversion provides more diffuse spatial information in these regions.

We find that the EDGAR v4.3.2 inventory prominently overestimates anthropogenic emissions over eastern China, likely from coal production, and around the Persian Gulf, likely from oil/gas production. The finding of a positive inventory bias in China is consistent with previous inversions of GOSAT data using EDGAR v4.2 or v4.1 as prior estimate (Monteil et al., 2013; Alexe et al., 2015; Turner et al., 2015; Pandey et al., 2016). We find that EDGAR underestimates emissions over Japan and Southeast Asia, where rice cultivation is the largest anthropogenic source but there are also large wetland emissions. There are



also large corrections in wetland areas of central Africa, South America, and North America.

We do not find large correction factors over the US, except for the southeastern coast which is likely due to an overestimate of methane emissions from coastal wetlands in the prior WetCHARTs inventory. This overestimate of US coastal wetland emissions in WetCHARTs is consistent with a previous inversion of aircraft observations over the Southeast US by Sheng et al. (2018b) and may be explained by low soil organic carbon in these ecosystems (Holmquist et al., 2018) and/or the overestimated impacts of partial wetland land-cover classes predominant in the southeastern US (Lehner and Döll, 2004; Bloom et al., 2017). Previous inversions found factor of 2 underestimates of EDGAR v4.2 emissions in the South-Central US (Miller et al., 2013; Turner et al., 2015) but we do not find such an underestimate here and attribute this to our use of the gridded version of the US EPA inventory as prior estimate (Maasakkers et al., 2016). EDGAR v4.2 allocated oil/gas emissions mainly according to population, which greatly underestimates emissions in oil/gas production regions in the South-Central US (Maasakkers et al., 2016).

Improved estimates of global methane emissions for the individual source types of Table 1 can be inferred from our results by assuming that the relative contributions from different source types in a given $4° × 5°$ grid cell is correct in the prior inventory. The global posterior estimate for a given source type is then obtained by applying the $4° × 5°$ posterior/prior ratios from Figure 4 to the distribution of source types in Figure 2. Results in Figure 5 indicate little change to the global prior inventory by source type even though there are large regional reallocations. Coal mining emissions decrease by 29% mainly due to China, and rice cultivation and livestock increase by 15% and 8% respectively, mainly driven by the tropics.

There has been particular interest in quantifying emissions from oil/gas exploitation because of the potential for large reductions of these emissions through simple control measures (Zavala-Araiza et al., 2015; Alvarez et al., 2018). The EDGAR v4.3.2 national oil/gas emission totals can differ greatly from the national (spatially unresolved) totals reported by individual countries to the United Nations Framework Convention on Climate Change (UNFCCC, 2017). This is shown in Figure 6 with national oil/gas emissions from the top ten countries in either the EDGAR v4.3.2 or UNFCCC inventories. We can estimate national oil/gas emission totals from our inversion by again assuming that the relative contributions of oil/gas to total emissions in individual $4° × 5°$ grid cells are correct, and by further mapping the $4° × 5°$ correction factors to the $0.1° × 0.1°$ emission EDGAR grid. The emission-weighted scaling factor is then used with the national oil/gas totals reported by EDGAR. Russia is the largest national source but the inversion is limited in its ability to constrain oil/gas emissions there because a third of these emissions are north of $60°N$ in EDGAR v4.3.2 (Figure 2).

Results in Figure 6 show that the inversion generally pushes the prior EDGAR v4.3.2 estimates of oil/gas emissions toward the UNFCCC values. One would expect the UNFCCC national reports to provide better estimates than EDGAR v4.3.2 because of their use of local information (Scarpelli et al., 2018) as compared to the more generic estimates (IPCC, 2006) used by EDGAR, similar to the IPCC Tier 1 methodology and using global datasets. Thus we find that EDGARv4.3.2 greatly underestimates emissions in Uzbekistan, which are high because of leaky infrastructure (Scarpelli et al., 2018). For Iran, Algeria,





Nigeria, Saudi Arabia, and Qatar we find much lower emissions than EDGAR v4.3.2 and more consistent with the UNFCCC data. For China we are in better agreement with EDGAR v4.3.2 than with the UNFCCC estimate, which relies on anomalously low emission factors (Larsen et al., 2015). In Venezuela we find higher emissions than either EDGAR v4.3.2 or UNFCCC. The latest available report from Venezuela to the UNFCCC dates back to 1999.

## 3.2 Spatial distribution and source attribution of methane emission trends

Figure 7 shows base inversion results for the linear emission trends on the $4° \times 5°$ grid for 2010-2015 and the associated averaging kernel sensitivities. Also shown in the bottom right panel is the 2010-2015 time series of posterior OH concentrations with error standard deviations from the posterior error covariance matrix. We find no significant OH trend over the period

10 although uncertainties are large. The information on the spatial distribution of emission trends originates from local/regional gradients of atmospheric methane observed by GOSAT, and we find from the posterior error covariance matrix of the inversion that it is not correlated with information on OH concentrations. Thus the large posterior uncertainty in global OH concentrations does not induce any significant correlated error in the spatial distribution of emission trends. This may be expected in view of the long lifetime of methane relative to the relevant time scales for atmospheric transport.

The GOSAT data provide seven independent pieces of information (DOFS) on the spatial distribution of the emission trend. Again, a Bayesian inversion without correcting for overfit ($\gamma = 1$) would erroneously indicate much higher DOFS. We find increasing emissions in the tropics and little change at higher latitudes. There are well-defined anthropogenic positive trends over China, India, and the Persian Gulf. Trends in China are in areas with dominant emissions from coal mining but also

significant contributions from livestock and waste. Trends over India are in areas of rice production but may also reflect waste management and livestock. The trend over India totals 0.4 (0.3-0.5) Tg a$^{-1}$ (range of the inversion ensemble). Ganesan et al. (2017) found a non-significant trend ($0.2 \pm 0.7$ Tg a$^{-1}$) over India for 2010-2015 using an ensemble of GOSAT, commercial aircraft (CARIBIC), and surface station methane data, but this does not exclude the significant increase that we find here. The trend over the US is less well defined and not well constrained but suggests an increase over the eastern part of the country

where multiple source types could contribute (Sheng et al., 2018a, b).

The bottom left panel of Figure 7 shows the attribution of the global increasing trend in emissions to individual source types, following the same assumption that was used in Figure 5 to attribute emissions to source types. We further separate tropical and extratropical contributions. Boreal wetland trends cannot be constrained by our inversion effectively (no observations north of

30 60° N). 43% of the 5 Tg a$^{-1}$ global emission trend found in the inversion for 2010-2015 is driven by wetlands (mainly tropical), 16% by livestock, and 11% by oil/gas. No source type shows a global decrease. Our source attribution of the methane trend is consistent with isotopic evidence suggesting that the increase in methane over the past decade has been driven by biogenic sources (Nisbet et al., 2016; Schwietzke et al., 2016) including tropical wetlands (McNorton et al., 2016).





### 3.3 Global methane budget and trends

The previous sections showed that our inversion of the GOSAT data is able to provide relatively fine information on the spatial distribution of methane emissions (DOFS = 128) as well as some information on the spatial distribution of 2010-2015 emission trends (DOFS = 7). This information on the spatial distribution originates from local/regional gradients of atmospheric methane

observed by GOSAT. We now examine to what extent error correlations may limit our ability to independently quantify the global emission of methane, the global tropospheric OH concentrations, and their respective trends.

To analyze the constraints from the inversion on the global budget of methane, we collapse the inversion to the reduced 4-member global state vector of 2010-2015 mean values described in Section 2.6 (global methane emission, global emission

trend, global tropospheric OH concentration, global OH trend). We use normal errors for all state vector elements (using log-normal errors could bias the mean). Table 2 compares the prior and posterior values for this global budget. The uncertainty in global emissions and trends is likely underestimated because of the lack of prior error covariance assumed between the 1009 grid cells. The global mean tropospheric OH concentration is expressed in terms of the corresponding methane lifetime $\tau_{CH_4}^{OH}$. Figure 8 shows the averaging kernel rows for this reduced global state vector ($\mathbf{A_{red}}$ in Section 2.6), measuring the sensitivity

of the inversion results to the true values (diagonal terms) and the aliasing due to error correlations (off-diagonal terms). We find that the mean 2010-2015 global methane emission and OH concentration are strongly and independently constrained, with averaging kernel sensitivities near unity and little error correlation. On the other hand, there is strong negative error correlation between emission trends and OH trends, and the OH trend can only be weakly constrained. This is illustrated further in Figure 9 with the joint probability density function (pdf) plots of the posterior estimates, where the confidence levels measure the

probability of a given value and the tilts of the ellipses measure the error correlations.

A major implication of being able to constrain independently the global methane emission and the global OH concentration is that satellite observations of atmospheric methane can provide an independent proxy for quantifying the global mean tropospheric OH concentration. Our posterior estimate of the methane lifetime $\tau_{CH_4}^{OH}$ is 10.8 ± 0.4 years. It is strongly constrained

by the inversion, as shown by the averaging kernel sensitivity near unity, and is thus largely independent from the prior estimate of 10.6 ± 1.1 years. So far the main method for estimating global OH has been through the methyl chloroform budget (Prather et al., 2012), but this is becoming problematic as methyl chloroform concentrations decrease and previously minor potential sources like ocean outgassing may become significant (Wennberg et al., 2004; Liang et al., 2017). Satellite observations of methane could provide an alternative. Our inversion confirms the best estimate of global OH from the methyl chloroform

budget (Prather et al., 2012) but reduces its uncertainty from 10% to 4%. The magnitude of reduction may be overoptimistic because of the idealized treatment of error statistics and the assumption that the global 3-D OH distribution in the forward model is correct. Zhang et al. (2018) present a more thorough error analysis of this potential of methane satellite observations as proxy for global OH concentrations.





We find on the other hand that there is large error correlation between our estimates of global 2010-2015 emission trends and OH trends, and limited ability to constrain the OH trend. The posterior estimates for the 2010-2015 trends are +0.84 ± 0.04 % a$^{-1}$ for emissions and -0.2 ± 0.8 % a$^{-1}$ for OH. The joint pdf in Figure 9 illustrates the error correlation between the two. Another factor driving the 2010-2015 atmospheric methane trend is the initial imbalance in the 2010 budget, which we

can derive from the posterior estimates of the mean 2010-2015 budget imbalance and trends, and the interannual variability of wetlands emissions as represented by WetCHARTS. Figure 10 shows the contributions of these different terms to the observed 2010-2015 methane growth. 2010 was a relatively high year for tropical wetlands emissions according to WetCHARTS, which acts to dampen the overall trend. We can state with some confidence that increasing tropical emissions (Figure 7) made an important contribution to the 2010-2015 methane trend but any conclusion about the effect of an OH trend is highly uncertain

including in its sign.

**Table 2.** Global 2010-2015 methane budget[a]

|  | Prior | Posterior |
|---|---|---|
| Mean emission (Tg a$^{-1}$) | 548 ± 10 | 546 ± 2 |
| Emission trend (% a$^{-1}$) | 0 ± 0.1 | 0.84 ± 0.04 |
| Mean methane lifetime $\tau_{CH_4}^{OH}$ (a)[b] | 10.6 ± 1.1 | 10.8 ± 0.4 |
| OH trend (% a$^{-1}$) | 0 ± 0.8 | -0.2 ± 0.8 |

[a] From the inversion optimizing (1) mean 2010-2015 methane emissions on the 4° × 5° grid, (2) linear methane emission trends on that same grid, (3) global mean 2010-2015 tropospheric OH concentration, and (4) linear trend in global OH concentrations. Expected values and error standard deviations are shown. The prior estimates are described in Section 2.2. The posterior global emission and its trend are obtained by summing the contributions from all 4° × 5° grid cells, and the error standard deviations are computed accounting for posterior error correlation. Minor methane sinks totaling 61 Tg a$^{-1}$ are not optimized in the inversion.

[b] Methane lifetime against oxidation by tropospheric OH, computed as the ratio between the total atmospheric mass of methane (including the stratosphere) and the annual loss rate from oxidation by OH in the troposphere.

## 4 Conclusions

We used 2010-2015 observations of atmospheric methane columns from the GOSAT satellite instrument in a global inverse analysis to optimize a state vector including (1) mean 2010-2015 methane emissions on a 4° × 5° grid, (2) 2010-2015 emission

trends on that same grid, and (3) global mean tropospheric OH concentrations for individual years. Our work aimed to improve current understanding of global methane sources and the renewed growth in atmospheric methane over the past decade, and to





examine if satellite observations can independently constrain methane emissions and tropospheric OH, the main methane sink.

Our inversion used the GEOS-Chem chemical transport model as forward model to relate the state vector elements (1)-(3) to atmospheric methane columns. We fitted the model to the GOSAT observations by analytical solution of the Bayesian problem,

including construction of the full Jacobian matrix. The analytical solution provides closed-form characterization of errors and of the information content in the solution. This is critical for diagnosing the ability of the GOSAT observations to constrain emission trends and to achieve separate constraints on emissions and OH concentrations. It also allows us to easily generate an ensemble of inversions testing different assumptions. Analytical solution of the inverse problem generally requires normal prior error distributions but we show here that it can be readily extended to log-normal prior error distributions by using a

simple scaling of the original Jacobian matrix.

Our optimization of mean 2010-2015 methane emissions on the $4° × 5°$ grid, achieves 128 degrees of information for signal (DOFS), with strong constraints in source regions. The EDGAR v4.3.2 anthropogenic emission inventory taken as default anthropogenic prior estimate in the inversion is too high in China (coal emissions) and in the Middle East (oil/gas emissions).

Oil/gas national totals in EDGAR v4.3.2 can differ greatly from the values reported by individual countries to the United Nations Framework Convention on Climate Change (UNFCCC), and our inversion results are generally more consistent with the UNFCCC estimates. We find little correction to anthropogenic US emissions when a new gridded version of the US EPA greenhouse gas inventory is used as anthropogenic prior estimate. Previous inverse studies that relied on the EDGAR v4.2 inventory as prior found large underestimates of US emissions, but this may reflect errors in the spatial distribution of EDGAR

v4.2 oil/gas emissions.

Optimization of methane emission trends over the 2010-2015 period yields a DOFS of 7 on the $4° × 5°$ grid, meaning that only strong source regions can be constrained. We find that the 2010-2015 increasing trend in atmospheric methane is mostly due to increasing emissions rather than decreasing OH concentrations. Most of the increase is in tropical wetlands, India, and

China. Trends in North America and Europe are small. Our findings are consistent with isotopic constraints pointing to tropical biogenic sources as responsible for the renewed growth of methane over the past decade.

We further examined the ability of the GOSAT data to constrain the global methane emission and its trend over the 2010-2015 period independently from the global OH concentration and its trend. For this purpose we considered a reduced 4-component

state vector consisting of (1) the global mean methane emission for 2010-2015, (2) the global emission trend over that period, (3) the global mean OH concentration for 2010-2015, (4) the global OH trend over that period. (1) and (2) were obtained by collapsing the inverse solutions for emissions on the $4° × 5°$ grid, so that the distributions of emissions and their trends are still optimized. Results show that the global methane emission (546 Tg a$^{-1}$) can be constrained independently from the global OH concentration (atmospheric methane lifetime against oxidation by tropospheric OH of $10.8 ± 0.4$ years), with little

error correlation. This is because methane emissions and loss have different and separable signatures on atmospheric methane





columns. An important implication is that satellite observations of atmospheric methane can serve as a useful proxy for the global OH concentration. In contrast, we find that errors on the 2010-2015 OH trends are strongly correlated with the stronger signal from emission trends.

Satellite observations of atmospheric methane are expected to vastly improve in the near future with the launch of the TROPOMI instrument in October 2017, the advent of geostationary observations from the GeoCARB instrument to be launched in the early 2020s, and other instruments measuring methane on local to global scales (Jacob et al., 2016). Our work with the relatively sparse GOSAT data suggests that this future constellation of satellites will enable the mapping of emissions at fine scales. Satellite observations of methane could also provide an effective means for monitoring OH concentrations, replacing
methyl chloroform whose ability to serve as an OH proxy is declining.

## Appendix A:  Appendix A: Comparison of forward model simulations at $4° \times 5°$ and $2° \times 2.5°$ resolutions

Stanevich (2018) pointed out significant global meridional biases in the GEOS-Chem simulation of methane columns at $4° \times 5°$ resolution relative to $2° \times 2.5°$, and they argued that $2° \times 2.5°$ was much better to use in global inversions of methane
sources. However, we find that most of the difference between the two resolutions is in the stratosphere, which we correct following Equation 1. Figure A1 illustrates this point with the differences between the two resolutions averaged over latitudinal bands. Values are 2010-2015 means for the full column and for the tropospheric column only. There are large high-latitude biases for the total column but these are mainly in the stratosphere. The tropospheric bias is less than 5 ppb at all latitudes. Results for individual seasons are similar. Buchwitz et al. (2015) consider that biases below 10 ppb are acceptable in methane
inversions.

## Appendix B:  Appendix B: Sensitivity to seasonal bias in prior emission estimates

The GEOS-Chem forward model simulation using prior emission estimates shows a seasonal background bias relative to GOSAT observations, for which we applied a latitude-dependent correction (Section 2.3). This correction could mask a bias in
the seasonality of prior emissions. We conducted an additional inversion in which we did not apply this seasonal correction and instead optimized emissions for individual seasons with no prior error correlation between seasons. This brings the total size of the state vector up to 5052, which challenges the power of the GOSAT observations to provide independent constraints. As shown in Figure A2, the effective posterior/prior ratios found by summing the seasonal emissions are very similar to the ones from the base inversion. This indicates that the global pattern of scaling factors is not driven by corrections made to improve
the seasonal agreement between the model and GOSAT. The effective scaling factors are smaller in magnitude and smoother than the previous results because fewer observations are available per state vector element, resulting in smoothing error (Turner





and Jacob, 2015).

*Author contributions.* JDM and DJJ designed the study. JDM performed the analysis. JDM, MPS, HN, and MH performed simulations and supporting simulations/analysis. JDM, DJJ, MPS, TRS, HN, JXS, YZ, MH, AAB, KWB, and JRW discussed the results. AAB provided the
5   WetCHARTS emissions and supporting data. GJM provided guidance and supporting data on the EDGAR v4.3.2 emissions. RJP provided the GOSAT data and supporting guidance. JDM and DJJ wrote the paper and all authors provided input on the paper for revision before submission.

*Acknowledgements.* This research was funded by the NASA Earth Science Division. Part of this research was carried out at the Jet Propulsion Laboratory, California Institute of Technology, under a contract with the National Aeronautics and Space Administration. R. J. Parker is with
10  funded by the UK National Centre for Earth Observation (NCEO grant number: nceo020005) and the ESA Greenhouse Gas Climate Change Initiative (GHG-CCI). R. J. Parker thanks the Japanese Aerospace Exploration Agency, National Institute for Environmental Studies, and the Ministry of Environment for the GOSAT data and their continuous support as part of the Joint Research Agreement. This research used the ALICE High Performance Computing Facility at the University of Leicester for the GOSAT retrievals. TCCON data were obtained from the TCCON data archive, hosted by CaltechDATA - tccondata.org (Strong et al., 2017; Wunch et al., 2015; Morino et al., 2014b; Wennberg et al., 2016a, 2014b; Iraci et al., 2016a; Kivi et al., 2014; Blumenstock et al., 2014; Iraci et al., 2016b; De Mazière et al., 2014; Dubey et al., 2014a; Te et al., 2014; Dubey et al., 2014b; Kawakami et al., 2014; Morino et al., 2014a; Hase et al., 2014; Griffith et al., 2014b; Sherlock et al., 2014; Sussmann and Rettinger, 2014; Griffith et al., 2014a; Deutscher et al., 2014; Wennberg et al., 2016b, 2014a; Notholt et al., 2014; Warneke et al., 2014).



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





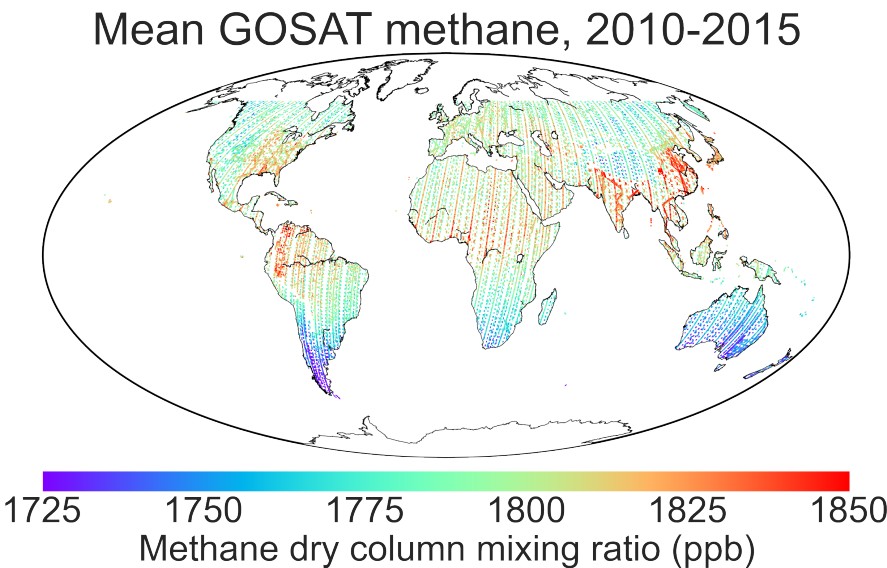

**Figure 1.** 2010-2015 average of the GOSAT methane dry column mixing ratios used in our inversion. Data are from the University of Leicester version 7 $CO_2$ proxy retrieval (Parker et al., 2015), excluding glint observations over the oceans and observations poleward of $60°$. GOSAT pixels are 10-km circular diameter and are inflated here to $0.5°$ for visibility. The red stripes are a sampling artefact as these retrievals are from towards the end of the time period.



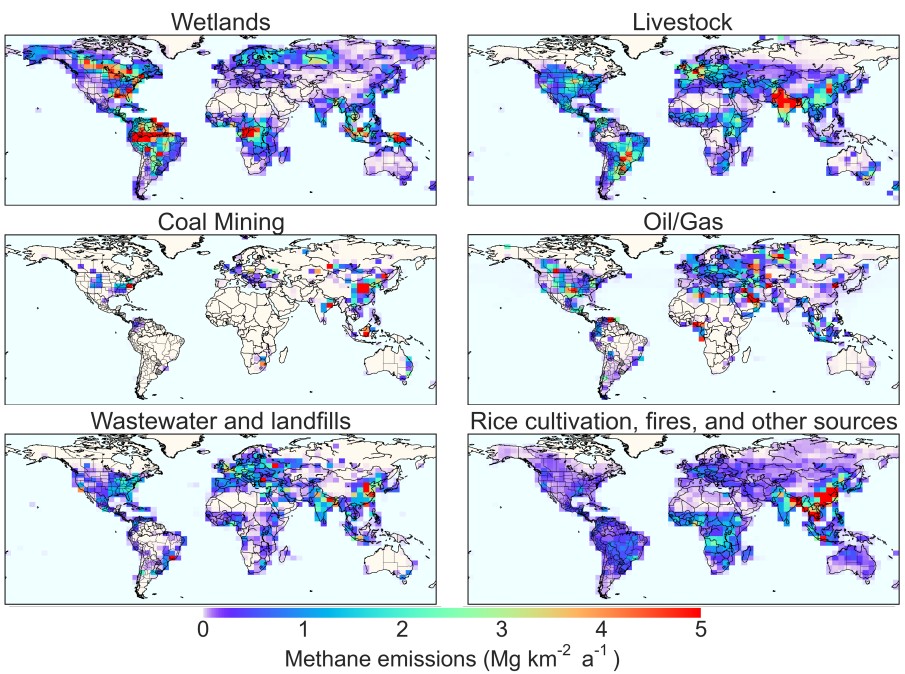

**Figure 2.** Prior estimates of methane emissions from wetlands, livestock, oil/gas, coal mining, wastewater and landfills, and other sources. Values are 2010-2015 averages and are shown on the $4° \times 5°$ GEOS-Chem grid used for the inversion.



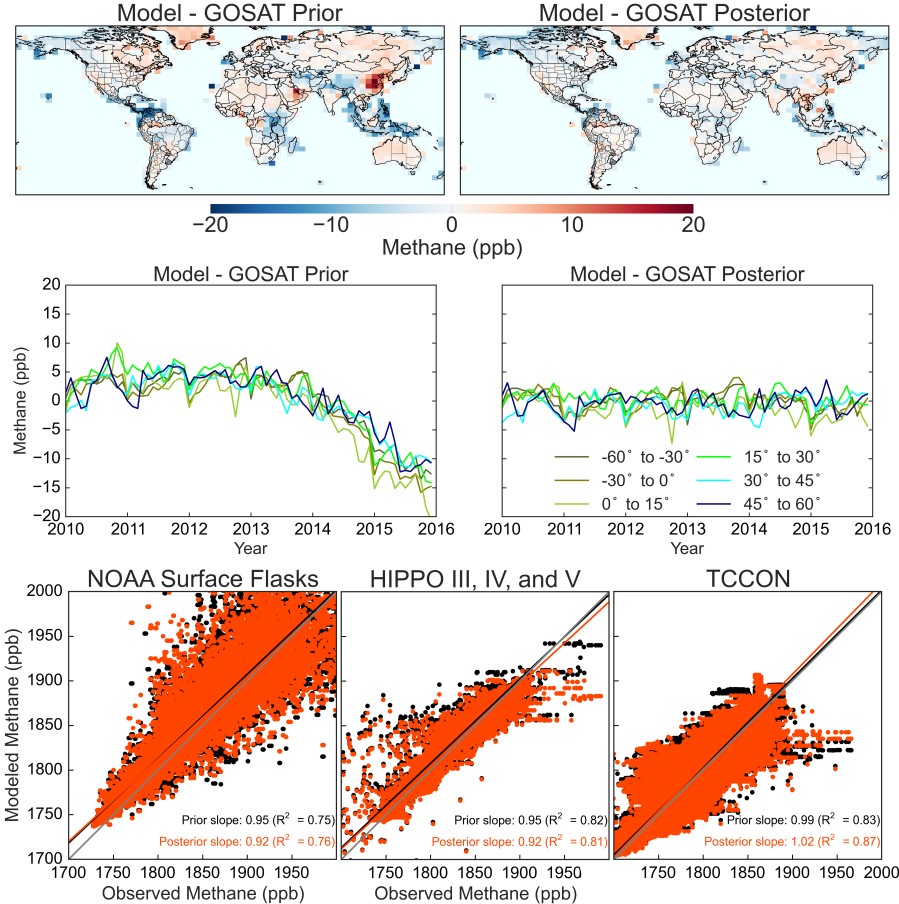

**Figure 3.** Comparisons of observed methane concentrations to the GEOS-Chem forward model using either prior or posterior (optimized) estimates of 2010-2015 emissions and OH concentrations. The top panels show mean 2010-2015 differences with GOSAT observations on the $4° \times 5°$ grid. The middle panels show the monthly time series of the differences averaged over latitude bands. The bottom panels show independent 2010-2015 comparisons to global observations from NOAA surface stations, HIPPO aircraft meridional cross-sections over the Pacific (2010 and 2011, with the model sampled along the flight tracks), and the TCCON network. Reduced major axis (RMA) regressions are as shown along with the 1:1 line (in grey). HIPPO observations are averaged over GEOS-Chem grid cells. The NOAA surface stations and HIPPO aircraft measure local methane dry air mole fractions while the TCCON network measures column-averaged dry-air mole factions.





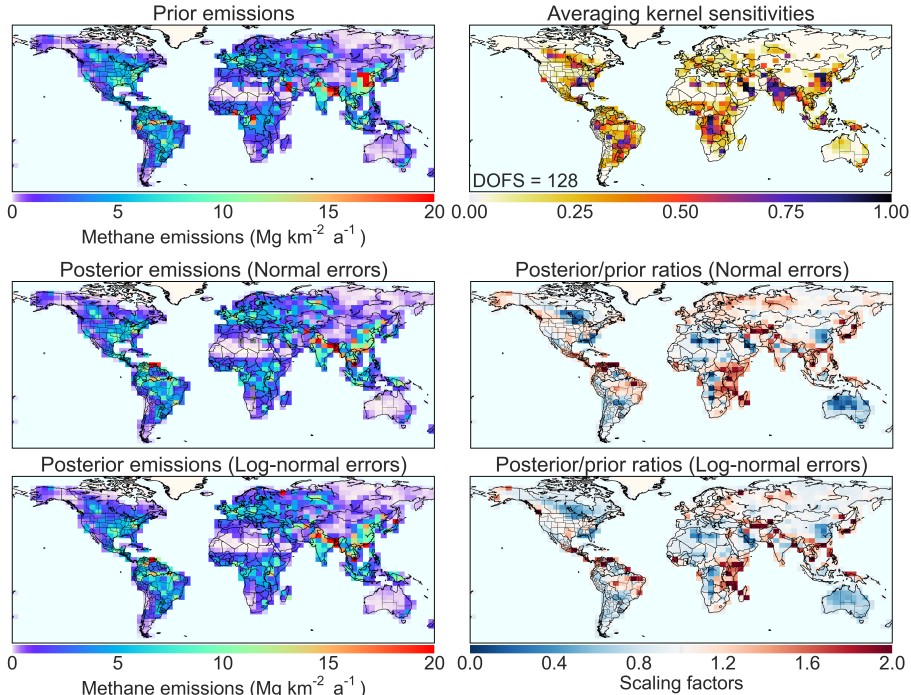

**Figure 4.** Optimization of the global distribution of mean 2010-2015 methane emissions using GOSAT observations. Prior emissions are in the top left panel (see breakdown in Figure 2). The top right panel shows averaging kernel sensitivities for the base inversion (diagonal elements of the averaging kernel matrix), with the degrees of freedom for signal (DOFS, trace of the averaging kernel matrix) in legend. The middle panels show the posterior emissions from the base inversion and the associated ratios between posterior and prior emissions. Grey grid cells (for example in North Africa and Australia) indicate small negative posterior emissions. The bottom row shows the same but for the inversion assuming log-normal prior errors, which does not allow for negative posterior emissions.




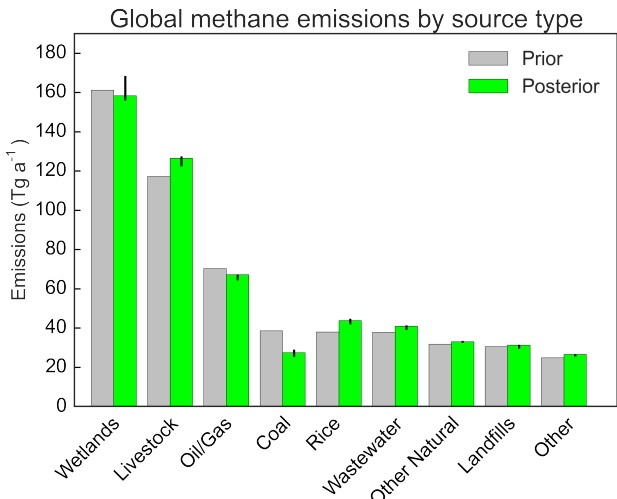

**Figure 5.** Global methane emissions by source type in the prior estimate for the inversion (Table 1) and in the posterior estimate. Values are 2010-2015 means. The attribution to source types in the posterior estimate is done by assuming that the relative contributions of different source types in individual $4° \times 5°$ grid cells are correct in the prior estimate. Posterior estimates are from the base inversion and error bars show the ranges of results from the inversion ensemble.



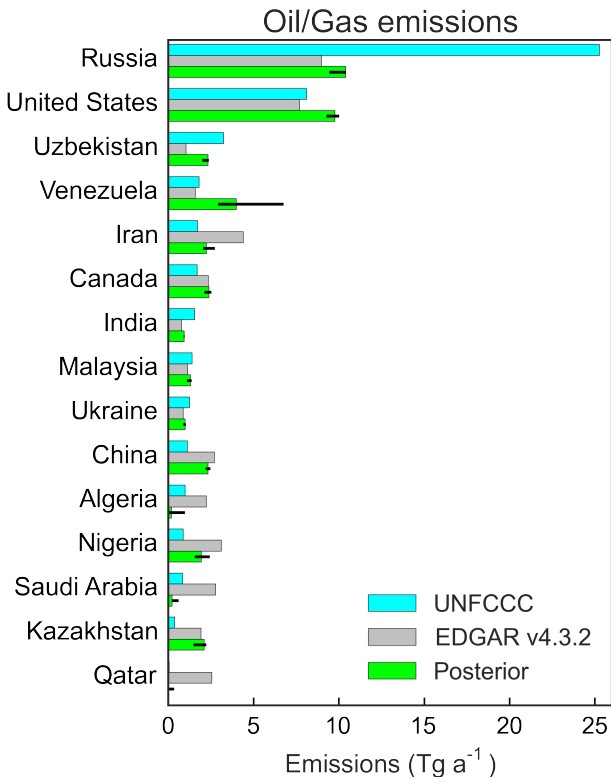

**Figure 6.** National estimates of methane emissions from the oil/gas industry. Values reported by individual countries to the UNFCCC for 2012 (Annex I countries) or the closest year (non-Annex I countries: Nigeria (1994), Venezuela (1999), Algeria (2000), Iran (2000), India (2010), Saudi Arabia (2010), and China (2012)) are compared to 2012 emissions from EDGAR v4.3.2 national oil/gas totals, and to the posterior values from our base inversion as described in the text. Black lines are ranges for the ensemble of inversions. Values are shown for the top ten emitting countries in either the UNFCCC or EDGAR v4.3.2 inventories. A large part of Russian emissions are too far north to be effectively constrained by the inversion.




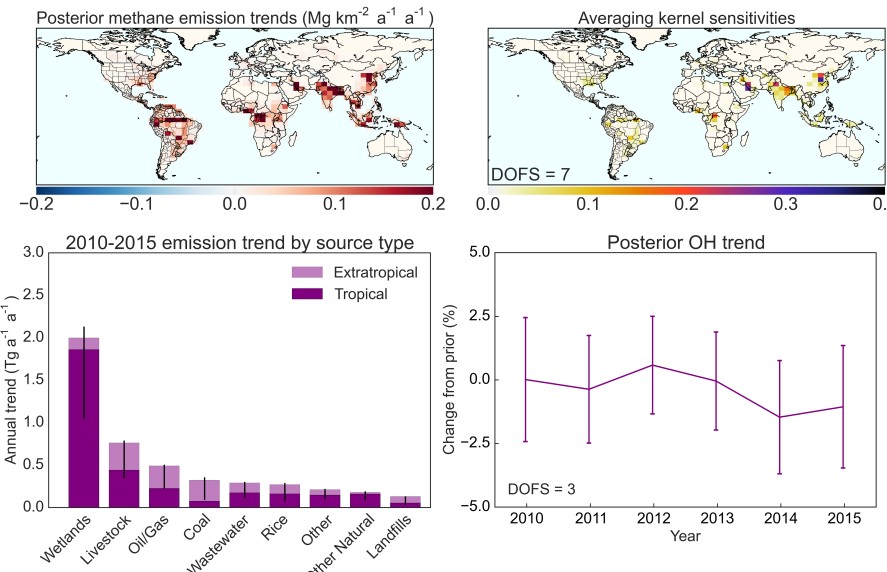

**Figure 7.** 2010-2015 methane emission trends and global tropospheric OH trends as optimized by the inversion of GOSAT data, and corresponding averaging kernel sensitivities (diagonal terms of the average kernel matrix). The degrees of information for signal or DOFS (trace of the averaging kernel matrix) is shown inset. The bottom left panel gives the global attribution of the emission trends to individual source types, with ranges from the inversion ensemble. Shaded sections of the bars indicate the contribution from the tropics (24°S-24°N). The vertical bars in the OH trend panel are the posterior error standard deviations from the base inversion. The 2010-2015 decreasing trend in OH concentrations is not statistically significant (95% confidence level).





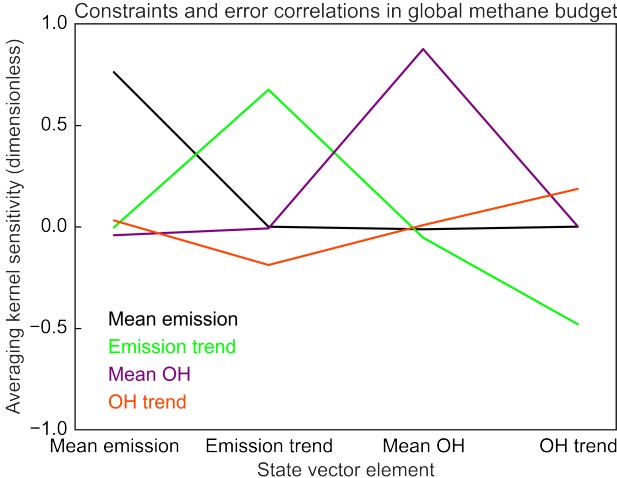

**Figure 8.** Constraints on the global 2010-2015 methane budget from our inversion of GOSAT data. The lines show the rows of the averaging kernel matrix $\mathbf{A_{red}}$ (Equation 9) for the reduced 4-element state vector consisting of the 2010-2015 mean emission, the linear emission trend, the 2010-2015 mean tropospheric OH concentration, and the linear OH trend.

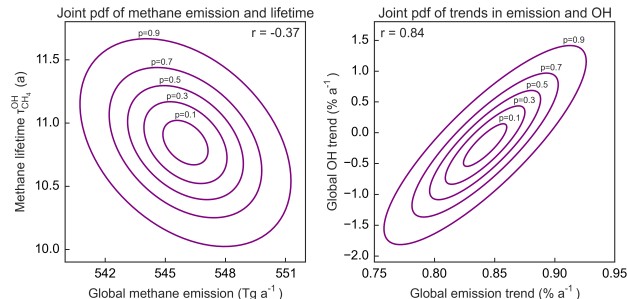

**Figure 9.** Joint probability density functions (pdfs) for the global methane budget as constrained by the 2010-2015 GOSAT data. The left panel shows the joint pdfs of the 2010-2015 global mean methane emission and methane lifetime against oxidation by tropospheric OH. The right panel shows the joint pdfs of the 2010-2015 global emission trend and OH trend. Contours show confidence ranges from 0.1 to 0.9. The error correlation coefficients are shown inset. The tilt of the ellipse indicates the extent of error correlation.



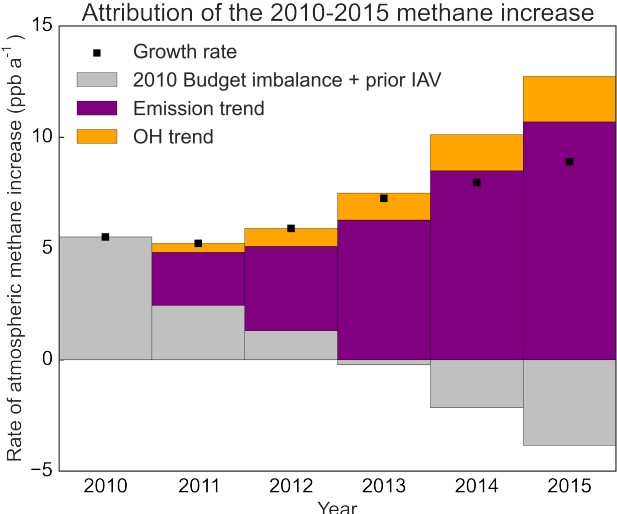

**Figure 10.** Attribution of the 2010-2015 increase in the atmospheric burden of methane. The grey bars show the trend imposed by the 2010 imbalance between sources and sinks combined with the interannual variability (IAV) of the prior estimate (mainly from wetlands). This trend decreases over the 2010-2015 period because the methane sink rises in response to the increasing concentration, and also because wetland emissions in 2010 are higher than in other years. Purple and orange show the contributions of the 2010-2015 methane emission trends and OH trends. The apportionment of the emission trend by source region and source type is shown in Figure 7. The OH trend has high uncertainty as discussed in the text.

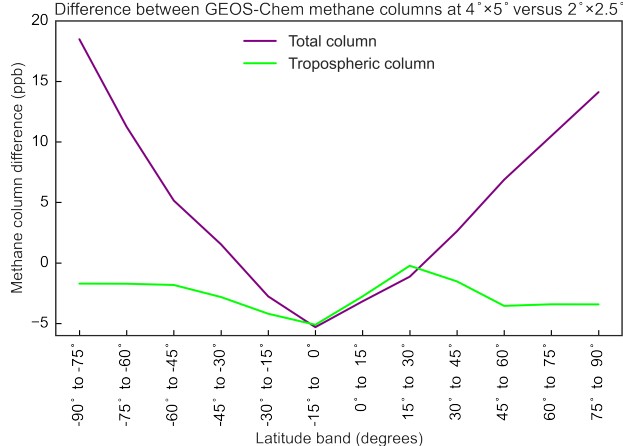

**Figure A1.** Difference between methane column concentrations simulated by GEOS-Chem at $4° \times 5°$ versus $2° \times 2.5°$. Values are 2010-2015 averages over latitudinal bands for total atmospheric columns and tropospheric columns.



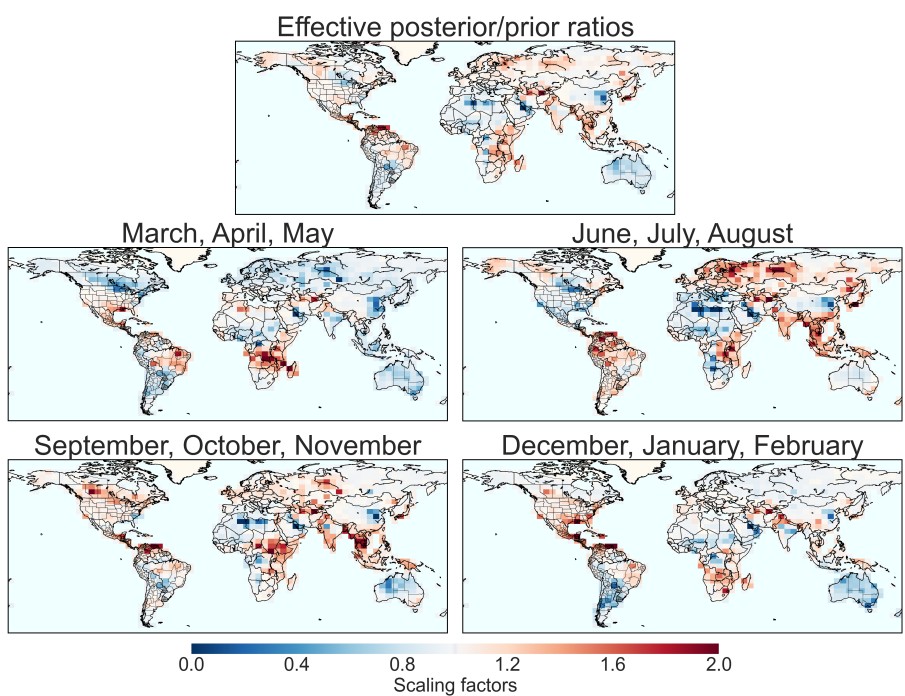

**Figure A2.** Results from the seasonal inversion, showing effective posterior scaling factors in the top panel and the seasonal scaling factors in the four bottom panels.