# Peer review of "Global distribution of methane emissions, emission trends, and OH concentrations and trends inferred from an inversion of GOSAT satellite data for 2010-2015"

_Atmospheric Chemistry and Physics, 2018_

## Referee Comment (RC1) · Anonymous Referee #1 · 8 Feb 2019

The manuscript "Global distribution of methane emissions, emission trends, and OH concentrations and trends inferred from an inversion of GOSAT satellite data for 2010-2015" from Maasakkers et al., submitted for publication in Atmos. Chem. Phys., describes the application of a methane emission and OH concentration inverse modelling scheme to six years of GOSAT column-averaged methane data to obtain information on methane emissions, OH concentrations and their trends. They explain the method including error characterization, the conducted sensitivity experiments and discuss the results, which are compared with published results and emission data bases. The pa-

per contains new results and covers aspects relevant for Atmos. Chem. Phys. The paper is very well written and I recommend publication after the mostly minor aspects listed below have been considered by the authors.

General comments

Several publications exist using GOSAT data in combination with inverse modelling to improve our knowledge of methane emissions. According to my knowledge, this is the first publication aiming at simultaneously inferring information on OH (which is the largest methane sink) together with methane emission information from satellite data.

Despite many efforts in recent year to obtain information on regional / country scale methane emissions and related trends from satellite data there are still many open questions (e.g., w.r.t. the reason or the reasons of the renewed methane growths after 2007) and the published results and conclusions are often apparently in contradiction. For example, in the abstract it is written: "The observed 2010-2015 growth in atmospheric methane is attributed mostly to an increase in emissions from India . . ." whereas the recent Nature communications article "Atmospheric observations show accurate reporting and little growth in India's methane emissions" (Ganesan et al., 2017, cited by the authors, and also using GOSAT satellite data) suggest that this seems not to be the case. Maasakkers et al. state (page 2, line 13 and following) that "Here we use global 2010-2015 methane observations from the GOSAT satellite in an analytical inverse analysis with full error characterization to better quantify methane sources and interpret the recent trend, including changes in both methane emissions and OH concentrations." The authors claim to use "full error characterization" and this is a bold statement. I recommend to formulate this less drastically. One aspect not "fully covered" are biases of the satellite data (e.g., unknown spatio-temporal error correlations). To assess the impact of biases is difficult but using an ensemble of satellite data should help to identify robust features and to obtain more robust conclusions. Here the authors used one GOSAT data product, namely the University of Leicester GHG-CCI v7 proxy product, but more products are available (from GHG-CCI and other

projects) and it is unclear to what extent the findings are related to the specific product used for this study. I am aware that using an ensemble of products would significantly increase the computer time and related resources as needs for data analysis etc. but I recommend to make use of the available ensemble of satellite products for future studies.

Specific comments

Page 3, line 20: The statement related to GOSAT observations (no drift and no degradation of data quality) citing Kuze et al., 2016, refers to the interferograms/spectra but not the derived methane product. Please modify that sentence to make this clear.

Page 6, line 24: Including a polynomial of latitude has (according to my knowledge) first been used by Bergamaschi et al. when applying methane inverse modelling to SCIAMACHY satellite data. I recommend to cite the relevant paper here.

Page 7, line 15: Sentence "SO is taken to be diagonal for lack of better information but the general effect of error correlation in the observations is accounted for in the inversion by a regularization factor". See above my "General comment". The "general effect" can be very different from the "specific effect" of biases of a given satellite product. The approach used may help or not to deal with this aspect. In the best case, it accounts for some aspects of error correlations but to what extent this is true is unclear.

Page 7, Eq. (2): Is the use of the gamma factor related to this specific inversion method or have similar methods to avoid overfitting also used by others (in their peer-reviewed publications)? If yes are the "scaling factors" similar? Reducing the weight of the observational terms by a factor of 20 seems very large. Would be good if additional information on this aspect could be added.

Page 13, line 23: Statement "but this does not exclude the significant increase that we find here". Why not? Please provide justification for this statement.

Page 29, Fig. 3: Middle right panel: What is the reason that the difference has a trend

only in 2014 and 2015? Please explain.

Page 29, Fig. 3: Bottom panels: The posterior slope is worse (larger deviation from 1.0). Is this a significant finding (or is the slope error too large)?

Page 35, caption Fig. 10: ". . . in response to the increasing concentration . . .". Which concentration? OH? Please add.

Page 35, Fig. 10: The shown growth rates appear to differ significantly from the growth rates published by NOAA (see https://www.esrl.noaa.gov/gmd/ccgg/trends_ch4/#global_growth). Can you comment on this?

Technical comments

Page 4, line 8: add space after "inventory".

Page 6, line 3: add space after "fields".

Page 11, line 3: add space after ".org".

Page 24, reference Sheng et al., 2018. The paper is published in ACP. Please update the reference.

---

## Referee Comment (RC2) · Anonymous Referee #2 · 9 Feb 2019

Methane's rising. Fast. We don't know why. What's happening to methane is arguably the most interesting current greenhouse problem, and there are very wide implications for efforts to mitigate climate warming. It makes the task of the Paris Agreement so much harder. Maasakkers et al discuss how best GOSAT satellite data can be used to tackle this very important problem, addressing the period 2010-2015, which includes years of very strong growth. Their approach is to use a global inverse analysis – from this not only do they obtain very interesting new estimates of methane emissions and growth trends, but also – a crucial question – they estimate the global abundance of

hydroxyl and its trend, and in so doing propose a new proxy for OH. The paper is very carefully written and it is thorough in its discussion of the methodological approaches. It takes note of the information available, both top down (e.g. using the NOAA data) and bottom up (inventory estimates). However, the study does give very short shrift to the C-isotopic constraints – although it is likely but not wholly demonstrated that the conclusions are compatible with the C-isotopic shift observed, the paper would be strengthened if the isotopic constraints were given a bit more discussion. Overall this paper is a major contribution and should be published with only minor revisions. It will be much cited.

Detailed comments that the authors may wish to consider.

Page 1 Line 14 Maybe give ±2 error on 546Tg. (e.g from table 2) P2 L4 "must"? – maybe "is likely to be". Also 'sources' – maybe better to say 'activities'. In general a 'source' is a thing like a cow, an emission has a flux, and an activity includes all manner of sins. P2 L5 No mention of the enormous number of tropical human-lit fires, from cow dung in India to grass and dead leaf fires in the Sahel?? P2 L8 Being picky – "concentration" is in a bucket of water. Mole fraction? Mixing ratio? Or here maybe the 'methane burden'. P2 L9 Being picky again - Citing the growth change in % is referring to a moving target. It would be much better to cite growth in parts per billion – i.e. not a % of a changing total. And does the text mean 1% of the whole burden or the anthropogenic increment? By saying 1% on Line 9, the text implies growth rates were around 16 to 17 ppb/yr in the 1980s – well, they were very high, but not that high, except perhaps in 1991. Also, 2014 growth was notably high: nearly 13 ppb in about 1820 – that's more than 0.4%. P2 L25 – GOSAT – maybe should mention Miller et al. Nature Communications 10, Article number: 303 (2019), either here or in the next paragraph. P4 L9 – 'aseasonal' – I'm not sure this is valid. Gas use in Eurasia at least is very winter-dependent, and gas pumping scheduled accordingly. Biomass burning in both the boreal realm (summer) and tropical (dry season) is very seasonal, and in the tropics is almost entirely anthropogenic – there are few lightning bolts in the

dry winters. P4 L13 – note that Petrenko et al.s work suggests the geological emissions are much smaller than previously suggested. Petrenko, V.V. et al. (2017) Minimal geological methane emissions during the Younger Dryas–Preboreal abrupt warming event. Nature doi:10.1038/nature23316 P4 L30 – 2.4% - quantify that in Tg and then the increase in ppb. A % is a moving thing. Also on P10 L33 P5 L30 – 'concentrations' again….and also on P5 L1 P6 L8 to 10 – Mention Naus, et al. (2019) Constraints and biases in a tropospheric two-box model of OH. Atmospheric Chemistry and Physics 19, 407-424.  and also perhaps Lelieveld, J., et al. (2016), Global tropospheric hydroxyl distribution, budget and reactivity, Atmos Chem Physics, 16, 12477-12493. P6 L 13 and P5 Table 1 – Cl sink of 9 Tg/yr – should mention Hossaini, R., et al. (2016) A global model of tropospheric chlorine chemistry: Organic versus inorganic sources and impact on methane oxidation.  Journal of Geophysical Research: Atmospheres 121.23 (2016). P6 L27-30 – the seasonal bias and correction – this is a weakness of the inputs and although it is comforting to know it doesn't affect the results significantly it should be target for future improvement. Seasonal fitting is a tool in identifying specific sources of emissions – wetlands emit less in drought; biomass burning is limited in the wettest periods.  So it's important to get seasonality right.  P7 L6 – s.d. 13 ppb – i.e about the same as growth in a strong-growth year.  P7 L30 – 'correlation in the observational error' maybe discuss this a little more – what is the 'regularization factor' allowing for? P8 L9 – ratio of elements 1009/7 – is this equalisation of OH and emissions correct : how about the prescribed non-OH sinks. P8 L15 – impact of methane change affecting OH – maybe it can be neglected globally but is that true at all latitudes (and longitudes)? P9 L4 and P 11 L16 – Soil uptake is probably quite strong in many moist well aerated wet tropical savanna woodlands, and also Cl uptake in the boundary layer may be strong in some locations. Thus negative emissions are likely in some areas.  P9 L19 – 'perviously' P9 L26 – significant sources in high northern latitudes and strongest OH in tropical troposphere.  P10 L22 – should probably discuss the Naus et al paper somewhere – maybe here? (see earlier comment on P6L8) P11 L2. Very interesting, especially overestimate in China. Maybe compare with Figure 1 in the Miller et al GOSAT paper – they see growth in China, India and tropical Africa. P11 L30 - 32 – in the East Asian context maybe mention Thompson et al, 2015 (Methane emissions in East Asia for 2000-2011 estimated using an atmospheric Bayesian inversion. Journal of Geophysical Research, 120)? Or 2018 (Variability in Atmospheric Methane From Fossil Fuel and Microbial Sources Over the Last Three Decades. Geophysical Research Letters, 45). P12 L17 – this finding on lower Chinese emissions is well substantiated and is a major conclusion that should be discussed in a bit more detail. P12 L18 – no mention of tropical wetland increases? Why are they excluded from likely sources of growth? Or tropical fires if there is a concurrent shift in another emission source that's masking the isotopic impact?? P12 L29 – absence of information north of 60N is a problem as many of Russia's largest gasfields are north of this line, and even in Canada and Norway there are gasfields are nearly at 60N. P12 L35 – Uzbek leaks – makes intuitive sense. P13 L 3 Likewise, intuitively Venezuela's industry is probably leaky and UNFCCC far from actuality. Note wetlands fringing Lake Maracaibo also likely a major source of quasi-natural emissions. P13 L18 – compare with Miller et al (2019) map? – China, India. P13 L20 – inside the error bounds of Ganesan et al, but sense is a bit different as this analysis suggests India is really quite strongly increasing. Should mention the surge in Indian coal production and general air pollution (India's GHG emisisons will soon surpass the European Union's). P13 L27 – source types. Here the paper should say more about the isotopic constraints and the various measurement-based papers by Schaefer et al, Nisbet et al and Schwietzke et al. There is a throw-way sentence at the end of the paragraph, but all the hypotheses about trends need to be consistent with the well-evidenced isotopic shift to lighter values. So far in the paper the discussion has been without isotopic constraint and that is OK, but only for the isotopes then to be used as an independent check on the inferences. For example the statement 'no source type shows a global decrease' – e.g. if biomass burning hasn't decreased a bit, it is then feasible but fairly constraining to make that statement tally with the hypothesis that a declining proportion of methane sourced from biomass burning is masking the isotopic impact of a fossil fuel increase?

P14 L16 – independence of CH4 and OH abundance constraints – this is very interesting and valuable if correct. P14 L24 – note that soil and Cl sinks are prescribed. P15 L3 and also P 16 L22– OH trend - rather different from favoured hypothesis in Turner et al (2017). P15 L 10 – any comment on the extraordinary 13 ppb growth in 2014? Table 2 State emission trend in ppb or in Tg/yr per year. 0.84±0.04% seems rather high compared to the NOAA record? P16 L33 – give error on 546 Tg/yr. P17 L17 – high lat biases in the stratosphere – interesting and may have isotopic impact. Fig 2 – would be good to have this large when typeset. Likewise Fig 3 top and Fig. 4. Fig 5 – anthropogenic biomass burning presumably fits in "other" – but this seems very small. African fires in particular are very large indeed and globally for example Saunois et al have the biomass burn & biofuel total more like 30-35Tg/yr. Fig 6 – Norway has a large gas industry. Australia has a large coal seam gas and also offshore gas industry. Turkmenia, UAE and Indonesia are all pretty big. Shouldn't they be on the chart? Much bigger producers than India for example. Fig. 10 – add some comment on the very high and global growth in 2014?
* * *

---

## Author Comment (AC1) · 27 Apr 2019

We thank the reviewers for their useful and knowledgeable comments that have improved our paper. Our responses and updated manuscript (with tracked changes) are included in the supplement.

Please also note the supplement to this comment:
https://www.atmos-chem-phys-discuss.net/acp-2018-1365/acp-2018-1365-AC1-supplement.pdf

---

## Author Response (AR2)

**We thank all reviewers for their thorough comments.**

**Reviewer 1**

The manuscript "Global distribution of methane emissions, emission trends, and OH concentrations and trends inferred from an inversion of GOSAT satellite data for 2010- 2015" from Maasakkers et al., submitted for publication in Atmos. Chem. Phys., describes the application of a methane emission and OH concentration inverse modelling scheme to six years of GOSAT column-averaged methane data to obtain information on methane emissions, OH concentrations and their trends. They explain the method including error characterization, the conducted sensitivity experiments and discuss the results, which are compared with published results and emission data bases. The paper contains new results and covers aspects relevant for Atmos. Chem. Phys. The paper is very well written and I recommend publication after the mostly minor aspects listed below have been considered by the authors.

**General comments**

Several publications exist using GOSAT data in combination with inverse modelling to improve our knowledge of methane emissions. According to my knowledge, this is the first publication aiming at simultaneously inferring information on OH (which is the largest methane sink) together with methane emission information from satellite data. Despite many efforts in recent year to obtain information on regional / country scale methane emissions and related trends from satellite data there are still many open questions (e.g., w.r.t. the reason or the reasons of the renewed methane growths after 2007) and the published results and conclusions are often apparently in contradiction. For example, in the abstract it is written: "The observed 2010-2015 growth in atmospheric methane is attributed mostly to an increase in emissions from India . . ." whereas the recent Nature communications article "Atmospheric observations show accurate reporting and little growth in India's methane emissions" (Ganesan et al., 2017, cited by the authors, and also using GOSAT satellite data) suggest that this seems not to be the case. Maasakkers et al. state (page 2, line 13 and following) that "Here we use global 2010-2015 methane observations from the GOSAT satellite in an analytical inverse analysis with full error characterization to better quantify methane sources and interpret the recent trend, including changes in both methane emissions and OH concentrations." The authors claim to use "full error characterization" and this is a bold statement. I recommend to formulate this less drastically.

**Full error characterization mainly refers to the error characterization coming from the analytical inversion compared to using an adjoint. We clarified this by now referring to it as "**closed-form error characterization**" everywhere.**

One aspect not "fully covered" are biases of the satellite data (e.g., unknown spatiotemporal error correlations). To assess the impact of biases is difficult but using an ensemble of satellite data should help to identify robust features and to obtain more robust conclusions. Here the authors used one GOSAT data product, namely the University of Leicester GHG-CCI v7 proxy product, but more products are available (from GHG-CCI and other projects) and it is unclear to what extent the findings are related to the specific product used for this study. I am aware that using an ensemble of products would significantly increase the computer time and related resources as needs for data analysis etc. but I recommend to make use of the available ensemble of satellite products for future studies.

Other GOSAT data products are consistent, we added a note on that in the GOSAT paragraph: "Other retrievals of GOSAT data are consistent with the University of Leicester product (Buchwitz et al., 2015)." Using ensembles of satellites are a great suggestion for future work but outside the scope of the current paper.

Specific comments

Page 3, line 20: The statement related to GOSAT observations (no drift and no degradation of data quality) citing Kuze et al., 2016, refers to the interferograms/spectra but not the derived methane product. Please modify that sentence to make this clear.

**Changed to incorporate that: "GOSAT **spectra** have shown no significant drift or degradation of data quality since the beginning of the record (Kuze et al., 2016)."**

Page 6, line 24: Including a polynomial of latitude has (according to my knowledge) first been used by Bergamaschi et al. when applying methane inverse modelling to SCIAMACHY satellite data. I recommend to cite the relevant paper here.

Added that citation: "This bias likely reflects a model overestimate of methane in the extratropical stratosphere (Saad et al., 2016), which is common across global models due to excessive meridional transport in the stratosphere (Patra et al., 2011) and was first seen in a SCIAMACHY inversion using the TM5 chemical transport model (Bergamaschi et al., 2007)."

Page 7, line 15: Sentence "SO is taken to be diagonal for lack of better information but the general effect of error correlation in the observations is accounted for in the inversion by a regularization factor". See above my "General comment". The "general effect" can be very different from the "specific effect" of biases of a given satellite product. The approach used may help or not to deal with this aspect. In the best case, it accounts for some aspects of error correlations but to what extent this is true is unclear.

**We improved the description of the regularization factor (see comment Page 7, Eq. (2)).**

Page 7, Eq. (2): Is the use of the gamma factor related to this specific inversion method or have similar methods to avoid overfitting also used by others (in their peer-reviewed publications)? If yes are the "scaling factors" similar? Reducing the weight of the observational terms by a factor of 20 seems very large. Would be good if additional information on this aspect could be added.

We improved the description of the regularization factor: "The variances in SO are underestimated because of correlation in the observational error that is missing in the diagonal formulation of SO and is difficult to quantify. We use  $\gamma$  to scale the original diagonal SO to get an optimal covariance matrix to be used in the inversion. Zhang et al. (2018) showed in an observing system simulation experiment (OSSE) for inversion of methane satellite data that a regularization factor  $\gamma = 0.05$  adjusts the variances optimally and prevents over- fitting. This was done by calculating the likelihood at x for a range of values of  $\gamma$ . Diagnosis of overfit and optimization of  $\gamma$  is readily done in an OSSE such as in Zhang et al. (2018) where the "true" solution is known. Here we find that using  $\gamma = 1$  (as in the pure Bayesian statement of the optimization problem) produces checkerboard patterns in the solution that are likely spurious. We choose  $\gamma = 0.05$  consistent with Zhang et al. (2018) for our base inversion as providing the best balance between prior and observational terms in the posterior value of the cost function. We examine the sensitivity to the choice of  $\gamma$  by conducting a sensitivity inversion with  $\gamma = 0.1$ ."

Page 13, line 23: Statement "but this does not exclude the significant increase that we find here". Why not? Please provide justification for this statement.

Clarified: "The trend over India totals 0.4 (0.3-0.5) Tg a-1 (range of the inversion ensemble). Ganesan et al. (2017) found a non-significant trend ( $0.2 \pm 0.7$  Tg a-1) over India for 2010-2015 using an ensemble of GOSAT, commercial aircraft (CARIBIC), and surface station methane data, **but our estimate is within their range**."

Page 29, Fig. 3: Middle right panel: What is the reason that the difference has a trend only in 2014 and 2015? Please explain.

We added additional explanation on this in the text: "The inversion corrects prior underestimates over tropical regions and an overestimate over China. It also fits the observed 2010-2015 trend in methane concentrations and its latitudinal distribution while the prior model underestimated the growth rate especially in 2014-2015."

Page 29, Fig. 3: Bottom panels: The posterior slope is worse (larger deviation from 1.0). Is this a significant finding (or is the slope error too large)?

The difference is small because the background is already well represented by the prior simulation. Slope errors are not shown here as they are arbitrarily small because a large number of datapoints originate from the same locations and are as such not fully uncorrelated.

Page 35, caption Fig. 10: ". . . in response to the increasing concentration . . .". Which concentration? OH? Please add.

We clarified that we are referring to the methane concentration: "This trend decreases over the 2010-2015 period because the methane sink rises in response to the increasing **methane** concentration, and also because wetland emissions in 2010 are higher than in other years."

Page 35, Fig. 10: The shown growth rates appear to differ significantly from the growth rates published by NOAA (see https://www.esrl.noaa.gov/gmd/ccgg/trends\_ch4/#global\_growth). Can you comment on this?

Added a comment on this in the associated text: "Our 2010-2015 growth rate averages 6.8, compared to 7.3 ppb  $a^{-1}$  in the NOAA record (esrl.noaa.gov/gmd/ccgg/trends\_ch4). The increase in the NOAA record is higher because of especially strong growth in 2014 (12.8 ppb) which is not fully captured by the linearized optimization used here. In our base inversion, this anomaly is explained by a reduced sink from OH."

Technical comments Page 4, line 8: add space after "inventory". Page 6, line 3: add space after "fields". Page 11, line 3: add space after ".org". Page 24, reference Sheng et al., 2018. The paper is published in ACP. Please update the reference.

We incorporated all of these comments.

**Reviewer 2**

Methane's rising. Fast. We don't know why. What's happening to methane is arguably the most interesting current greenhouse problem, and there are very wide implications for efforts to mitigate climate warming. It makes the task of the Paris Agreement so much harder. Maasakkers et al discuss how best GOSAT satellite data can be used to tackle this very important problem, addressing the period 2010-2015, which includes years of very strong growth. Their approach is to use a global inverse analysis – from this not only do they obtain very interesting new estimates of methane emissions and growth trends, but also – a crucial question – they estimate the global abundance of hydroxyl and its trend, and in so doing propose a new proxy for OH. The paper is very carefully written and it is thorough in its discussion of the methodological approaches. It takes note of the information available, both top down (e.g. using the NOAA data) and bottom up (inventory estimates). However, the study does give very short shrift to the C-isotopic constraints – although it is likely but not wholly demonstrated that the conclusions are compatible with the C-isotopic shift observed, the paper would be strengthened if the isotopic constraints were given a bit more discussion. Overall this paper is a major contribution and should be published with only minor revisions. It will be much cited.

Detailed comments that the authors may wish to consider. Page 1 Line 14 Maybe give  $\pm 2$  error on 546Tg. (e.g from table 2)

Because the uncertainty in global emissions is likely underestimated because of the lack of prior error covariance assumed between the grid cells we opt not to show the uncertainty without that explanation in the abstract we did include it in the conclusion (see comment P16 L33).

P2 L4 "must"? – maybe "is likely to be". Also 'sources' – maybe better to say 'activities'. In general a 'source' is a thing like a cow, an emission has a flux, and an activity includes all manner of sins.

**Changed to: "is most likely to be" and "anthropogenic activities".**

P2 L5 No mention of the enormous number of tropical human-lit fires, from cow dung in India to grass and dead leaf fires in the Sahel??

**Included biomass burning: "...anthropogenic activities including the oil/gas industry, coal mining, livestock, landfills, wastewater treatment, **biomass burning**, and rice cultivation"**

P2 L8 Being picky – "concentration" is in a bucket of water. Mole fraction? Mixing ratio? Or here maybe the 'methane burden'.

**Changed to: "The methane burden rose by"**

P2 L9 Being picky again - Citing the growth change in % is referring to a moving target. It would be much better to cite growth in parts per billion – i.e. not a % of a changing total. And does the text mean 1% of the whole burden or the anthropogenic increment? By saying 1% on Line 9, the text implies growth rates were around 16 to 17 ppb/yr in the 1980s – well, they were very high, but not that high, except perhaps

in 1991. Also, 2014 growth was notably high: nearly 13 ppb in about 1820 -that's more than 0.4%.

Changed all increases to average concentration per year: "The methane burden rose by ~ 12 ppb  $a^{-1}$  in the late 80s and by ~ 6 ppb  $a^{-1}$  in the 90s, plateaued in the early 2000s (~ 0.5 ppb  $a^{-1}$ ), and has resumed increasing at ~ 7 ppb  $a^{-1}$  since 2007 (esrl.noaa.gov/gmd/ccgg/trends\_ch4), for reasons that remain unclear (Turner et al., 2017)."

P2 L25 – GOSAT – maybe should mention Miller et al. Nature Communications 10, Article number: 303 (2019), either here or in the next paragraph.

Added Miller et al. to the list of GOSAT inversions, clarified that we optimize emissions and trends globally: "A number of inverse analyses have used the GOSAT data to improve estimates of methane emissions (Monteil et al., 2013; Cressot et al., 2014; Alexe et al., 2015; Turner et al., 2015; Pandey et al., 2016, 2017; **Miller et al., 2019**). Here we use the GOSAT data to optimize not only **global** emissions but also their 2010-2015 trends together with OH concentrations and their trends."

P4 L9 – 'aseasonal' – I'm not sure this is valid. Gas use in Eurasia at least is very winter-dependent, and gas pumping scheduled accordingly. Biomass burning in both the boreal realm (summer) and tropical (dry season) is very seasonal, and in the tropics is almost entirely anthropogenic – there are few lightning bolts in the dry winters.

Added that there is no better prior information available at this point: "Anthropogenic emissions are **assumed as aseasonal for lack of better prior information** except for manure management and rice cultivation."

P4 L13 – note that Petrenko et al.s work suggests the geological emissions are much smaller than previously suggested. Petrenko, V.V. et al. (2017) Minimal geological methane emissions during the Younger Dryas–Preboreal abrupt warming event. Nature doi:10.1038/nature23316

Added note: "While global geological emissions have previously been estimated to be over 50 Tg  $a^{-1}$  (Kirschke et al., 2013), Petrenko et al. (2017) showed that based on ice core measurements they should no higher than 15 Tg  $a^{-1}$ ."

P4 L30 - 2.4% - quantify that in Tg and then the increase in ppb. A % is a moving thing. Also on P10 L33 P5 L30 - 'concentrations' again. . ..and also on P5 L1

Done: "Our global prior estimate of mean methane emissions for the 2010-2015 period exceeds the sinks by 13 Tg  $a^{-1}$  (Table 1), which drives a 5 ppb  $a^{-1}$  increase in methane concentrations over that period even in the absence of an emission trend." To maintain the flow of the paper, we left most 'concentrations' in.

P6 L8 to 10 Mention Naus, et al. (2019) Constraints and biases in a tropospheric twobox model of OH. Atmospheric Chemistry and Physics 19, 407-424. and also perhaps Lelieveld, J., et al. (2016), Global tropospheric hydroxyl distribution, budget and reactivity, Atmos Chem Physics, 16, 12477-12493. Added a reference to Naus et al. (2019) in the introduction where we discuss two-box models: "Turner et al. (2017) find from a global 2-box model analysis that the surface record of methane observations is too sparse to arbitrate between methane emissions and OH concentrations as drivers for the methane increase, though Naus et al. (2019) pointed out that there are inherent biases in the 2-box modeling approach."

To include a comparison to model-based OH like Lelieveld et al. (2016), we included a reference to the multi-model mean from Naik et al. in the section that discusses the methane lifetime: "The prior estimate of the global tropospheric OH concentration is based on a GEOS-Chem full-chemistry simulation (Wecht et al., 2014) that yields a methane lifetime  $\tau^{OH}_{CH4}$  of 10.6 years, consistent with the best estimate inferred from the methylchloroform proxy (Prather et al., 2012) and the 9.7 ± 1.5 years estimate from the ACCMIP model ensemble (Naik et al., 2013)."

P6 L 13 and P5 Table 1 – Cl sink of 9 Tg/yr – should mention Hossaini, R., et al. (2016) A global model of tropospheric chlorine chemistry: Organic versus inorganic sources and impact on methane oxidation. Journal of Geophysical Research: Atmospheres 121.23 (2016).

Added a sentence: "The loss from oxidation by Cl totals 9 Tg  $a^{-1}$ , intermediate between the 12-13 Tg  $a^{-1}$  estimated by Hossaini et al. (2016) using the TOMCAT chemical transport model and 5.3 Tg  $a^{-1}$  estimated by Wang et al. (2019) in a GEOS-Chem simulation with full accounting of tropospheric chlorine. These minor sinks are not optimized in the inversion."

P6 L27-30 – the seasonal bias and correction – this is a weakness of the inputs and although it is comforting to know it doesn't affect the results significantly it should be target for future improvement. Seasonal fitting is a tool in identifying specific sources of emissions – wetlands emit less in drought; biomass burning is limited in the wettest periods. So it's important to get seasonality right.

We agree, a seasonal correction that does not depend on the model-observations difference would be superior and a great suggestion for future work. However, the method used here still allows for some grid-level seasonal fitting because of the latitudinal form of the seasonal correction, the appendix shows that resulting emissions are similar with and without seasonal correction.

P7 L6 - s.d. 13 ppb - i.e about the same as growth in a strong-growth year.

While we agree that is a good comparison in terms of order of magnitude we though it would be confusing to add here since the standard deviation concerns individual retrievals and not annual averages.

P7 L30 - 'correlation in the observational error' maybe discuss this a little more – what is the 'regularization factor' allowing for?

We improved the description of the regularization factor: "The variances in SO are underestimated because of correlation in the observational error that is missing in the diagonal formulation of SO and is difficult to quantify. We use  $\gamma$  to scale the original diagonal SO to get an optimal covariance matrix to be used in the inversion. Zhang et al. (2018) showed in an observing system simulation experiment (OSSE) for inversion of methane satellite data that a regularization factor  $\gamma = 0.05$  adjusts the variances optimally and prevents over- fitting. This was done by calculating the likelihood at x for a range of values of  $\gamma$ . Diagnosis of overfit and optimization of  $\gamma$  is readily done in an OSSE such as in Zhang et al. (2018) where the "true" solution is known. Here we find that using  $\gamma = 1$  (as in the pure Bayesian statement of the optimization problem) produces checkerboard patterns in the solution that are likely spurious. We choose  $\gamma = 0.05$  consistent with Zhang et al. (2018) for our base inversion as providing the best balance between prior and observational terms in the posterior value of the cost function. We examine the sensitivity to the choice of  $\gamma$  by conducting a sensitivity inversion with  $\gamma = 0.1$ ."

P8 L9 – ratio of elements 1009/7 – is this equalisation of OH and emissions correct : how about the prescribed non-OH sinks.

Minor sinks are not optimized in the inversion as mentioned in paragraph 2.3. Added that the equalisation ensures that there is no cost-function biased towards OH or emissions: "To provide equal weight to OH and emissions for explaining global methane trends, we increase the weight of the OH terms in the cost function (through the OH components of  $S_a$ ) by the ratio of the number of state vector elements 1009/7 so that from a cost-function perspective, a change in OH and global methane emissions are equally expensive. The sensitivity inversion assuming 10% prior error standard deviation on OH instead of 3% is equivalent to decreasing this weighting by a factor of 11."

P8 L15 – impact of methane change affecting OH – maybe it can be neglected globally but is that true at all latitudes (and longitudes)?

We added that methane is well-mixed so changes in local concentrations are relatively small as well: "There is a small non-linearity from the optimization of OH concentrations because changes in the methane concentrations affect the loss rate (Houweling et al., 2017) which we neglect because changes in methane concentrations are small **and methane is well-mixed globally.**"

P9 L4 and P 11 L16 – Soil uptake is probably quite strong in many moist well aerated wet tropical savanna woodlands, and also Cl uptake in the boundary layer may be strong in some locations. Thus negative emissions are likely in some areas.

Added Cl uptake on P9 L4 and rephrased: "Negative emissions could conceivably be attributed to locally strong soil uptake or oxidation by Cl atoms, but may also be unphysical (Miller et al., 2014)."

P9 L19 – 'perviously'

Changed to "previously".

P9 L26 – significant sources in high northern latitudes and strongest OH in tropical troposphere.

Clarified we mean the spatial/seasonal imprint: "Some separation is expected because sources of methane have a different **spatial/seasonal** imprint on the global methane distribution than the OH sink..."

P10 L22 – should probably discuss the Naus et al paper somewhere – maybe here? (see earlier comment on P6L8)

**Discussed Naus et al. 2019 at comment P6L8.**

P11 L2. Very interesting, especially overestimate in China. Maybe compare with Figure 1 in the Miller et al GOSAT paper – they see growth in China, India and tropical Africa.

Figure 3 shows 2010-2015 means, we clarified that in the caption: "The top panels show differences between model and GOSAT observations for 2010-2015 means on the  $4^{\circ} \times 5^{\circ}$  grid."

P11 L30 - 32 – in the East Asian context maybe mention Thompson et al, 2015 (Methane emissions in East Asia for 2000-2011 estimated using an atmospheric Bayesian inversion. Journal of Geophysical Research, 120)? Or 2018 (Variability in Atmospheric Methane From Fossil Fuel and Microbial Sources Over the Last Three Decades. Geophysical Research Letters, 45).

Added the 2015 Thompson paper in the description of mean emissions: "We find that the EDGAR v4.3.2 inventory prominently overestimates anthropogenic emissions over eastern China, likely from coal production, and around the Persian Gulf, likely from oil/gas production. The finding of an EDGAR overestimate in China is consistent with previous inversions of GOSAT data using EDGAR v4.1, v4.2, and v4.2FT2010 as prior estimate (Monteil et al., 2013; Thompson et al., 2015; Alexe et al., 2015; Turner et al., 2015; Pandey et al., 2016)."

Added the 2015 Thompson in the discussion on the trend as well: "There are well-defined anthropogenic positive trends over China, India, and the Persian Gulf. Trends in China are in areas with dominant emissions from coal mining but also significant contributions from livestock and waste. In an inversion of surface observations, Thompson et al. (2015) previously found an increasing trend over China for 2000-2011 which they attributed to coal mining."

Added a citation to Thompson 2018 in the intro section regarding trends: "A trend towards isotopically lighter methane has been attributed to an increase in microbial sources such as livestock and wetlands (Schaefer et al., 2016; Schwietzke et al., 2016; Nisbet et al., 2016; McNorton et al., 2016; Thompson et al., 2018). "

P12 L17 – this finding on lower Chinese emissions is well substantiated and is a major conclusion that should be discussed in a bit more detail.

Added a possible explanation: "The overestimate of coal mining emissions may be because standard IPCC emission factors used by EDGAR v4.2 were too high for Chinese coal mines and recovery of coal mine methane is not sufficiently taken into account (Peng et al., 2016). Emission factors were decreased in EDGAR v4.3.2 (Janssens-Maenhout et al., 2019) but we still find an overestimate."

P12 L18 – no mention of tropical wetland increases? Why are they excluded from likely sources of growth? Or tropical fires if there is a concurrent shift in another emission source that's masking the isotopic impact??

Clarified that this paragraph discussed average emissions, not trends: "Results in Figure 5 indicate little change to 2010-2015 average emissions compared to the global prior inventory by source type even though there are large regional reallocations."

P12 L29 – absence of information north of 60N is a problem as many of Russia's largest gasfields are north of this line, and even in Canada and Norway there are gasfields are nearly at 60N.

This is mentioned later on in the oil/gas section.

P12 L35 – Uzbek leaks – makes intuitive sense. P13 L 3 Likewise, intuitively Venezuela's industry is probably leaky and UNFCCC far from actuality. Note wetlands fringing Lake Maracaibo also likely a major source of quasi-natural emissions.

Corrections to wetlands in South America are noted in an earlier paragraph, here we mapped correction factors to oil/gas emissions specifically.

P13 L18 – compare with Miller et al (2019) map? – China, India.

Added that Miller found lower average emissions over China and trends over China and India:

"The finding of an EDGAR overestimate in China is consistent with previous global inversions of GOSAT data using EDGAR v4.1, v4.2, and v4.2FT2010 as prior estimate (Monteil et al., 2013; Thompson et al., 2015; Alexe et al., 2015; Turner et al., 2015; Pandey et al., 2016) and a regional inversion using EDGAR v4.3.2 (Miller et al., 2019)."

"In an inversion of surface observations, Thompson et al. (2015) previously found an increasing trend over China for 2000-2011 which they attributed to coal mining. Miller et al. (2019) found that this trend continued up to 2015 using GOSAT in a regional inversion."

"The trend over India is 0.4 (0.3-0.5) Tg  $a^{-1}$  (range of the inversion ensemble), consistent with the 2010-2015 trend of 0.7 ± 0.5 Tg  $a^{-1}$  trend from a regional GOSAT inversion by Miller et al. (2019)."

P13 L20 – inside the error bounds of Ganesan et al, but sense is a bit different as this analysis suggests India is really quite strongly increasing. Should mention the surge in Indian coal production and general air pollution (India's GHG emisisons will soon surpass the European Union's).

Improved the wording of the comparison with Ganesan and added a line on what EDGAR predicts based on activity data: "Ganesan et al. (2017) found a non-significant trend ( $0.2 \pm 0.7$  Tg a-1) over India for 2010-2015 using an ensemble of GOSAT, commercial aircraft (CARIBIC), and surface station methane data, but our estimate is not incompatible their range. EDGAR v4.3.2 predicts a 0.4 Tg a-1 increase in anthropogenic emissions from India between 2010 and 2012 mainly from livestock, coal, and waste based on increasing activity data (this trend is not included in our prior)"

P13 L27 – source types. Here the paper should say more about the isotopic constraints and the various measurement-based papers by Schaefer et al, Nisbet et al and Schwietzke et al. There is a throw-way sentence at the end of the paragraph, but all the hypotheses about trends need to be consistent with the well-evidenced isotopic shift to lighter values. So far in the paper the discussion has been without isotopic constraint and that is OK, but only for the isotopes then to be used as an independent check on the inferences. For example the statement 'no source type shows a global decrease' – e.g. if biomass burning hasn't decreased a bit, it is then feasible but fairly constraining to make that statement tally with the hypothesis that a declining proportion of methane sourced from biomass burning is masking the isotopic impact of a fossil fuel increase?

Added a discussion of biomass burning emissions: "Our source attribution of the methane trend is consistent with isotopic evidence suggesting that the increase in methane over the past decade has been driven by biogenic sources **outside the Arctic** (Nisbet et al., 2016; Schwietzke et al., 2016; Schaefer et al., 2016), including tropical wetlands (McNorton et al., 2016). Worden et al. (2017) previously found a decrease in biomass burning from 2001-2007 to 2008-2014 but no significant change for the 2010-2015 period investigated here. Their argument that a decrease in the biomass burning emissions would have masked the effect of an increase in fossil fuel emissions on the isotope signature of methane would not apply for our time period."

P14 L16 – independence of CH4 and OH abundance constraints – this is very interesting and valuable if correct. P14 L24 – note that soil and Cl sinks are prescribed.

Added: "The magnitude of reduction may be overoptimistic because of the idealized treatment of error statistics, the assumption that the global 3-D OH distribution in the forward model is correct, and the assumption that the minor sinks (Table 1) are correct."

P15 L3 and also P 16 L22– OH trend - rather different from favoured hypothesis in Turner et al (2017).

Turner et al. (2017) did not really present that as their favored hypothesis but rather as mathematically most likely outcome of the under-constrained problem in their box-model setup. Because of that we did not include a comparison here.

P15 L 10 – any comment on the extraordinary 13 ppb growth in 2014? Table 2 State emission trend in ppb or in Tg/yr per year.  $0.84\pm0.04\%$  seems rather high compared to the NOAA record?

Added Tg/yr numbers in the text: "Our posterior estimates for the 2010-2015 trends are  $+0.84 \pm 0.04 \% a^{-1}$  (4.6  $\pm$  0.2 Tg  $a^{-1}$ ) for emissions and  $-0.2 \pm 0.8 \% a^{-1}$  (-1.0  $\pm$  3.8 Tg  $a^{-1}$ ) for OH." These trends are for emissions and OH, a comparison of the concentration growth in our posterior simulation to NOAA is added below with an additional note on the 2014 growth: "Our 2010-2015 growth rate averages 6.8, compared to 7.3 ppb  $a^{-1}$  in the NOAA record (esrl.noaa.gov/gmd/ccgg/trends\_ch4). The increase in the NOAA record is higher because of especially strong growth in 2014 (12.8 ppb) which is not fully captured by the linearized optimization used here. In our base inversion, this anomaly is explained by a reduced sink from OH."

P16 L33 – give error on 546 Tg/yr.

We included the error estimate here.

P17 L17 – high lat biases in the stratosphere – interesting and may have isotopic impact.

**As mentioned in the text, the latitudinal bias is corrected before the inversion.**

Fig 2 – would be good to have this large when typeset. Likewise Fig 3 top and Fig. 4.

**All these figures are set as two-column figures.**

Fig 5 – anthropogenic biomass burning presumably fits in "other" – but this seems very small. African fires in particular are very large indeed and globally for example Saunois et al have the biomass burn & biofuel total more like 30-35Tg/yr.

**Added clarification from Table 1 here: "(Table 1, "other" includes fossil fuel combustion, industrial processes, and agricultural field burning)"**

Fig 6 – Norway has a large gas industry. Australia has a large coal seam gas and also offshore gas industry. Turkmenia, UAE and Indonesia are all pretty big. Shouldn't they be on the chart? Much bigger producers than India for example.

**As mentioned in the caption, we show countries in the top ten of either the EDGAR v4.3.2 or UNFCCC inventories.**

Fig. 10 – add some comment on the very high and global growth in 2014?

**Added a note on this in the associated text, see comment P15L10.**

**Global distribution of methane emissions, emission trends, and OH concentrations and trends inferred from an inversion of GOSAT satellite data for 2010-2015**

Joannes D. Maasakkers1,\*, Daniel J. Jacob1, Melissa P. Sulprizio1, Tia R. Scarpelli1, Hannah Nesser1, Jian-Xiong Sheng1, Yuzhong Zhang1, Monica Hersher1, A. Anthony Bloom2, Kevin W. Bowman2,3, John R. Worden2, Greet Janssens-Maenhout4, and Robert J. Parker5,6,7

1Harvard University, Cambridge, Massachusetts 02138, United States 2Jet Propulsion Laboratory, California Institute of Technology, Pasadena, CA, USA 3Joint Institute for Regional Earth System Science and Engineering, University of California, Los Angeles, CA, USA 4European Commission Joint Research Centre, Ispra (Va), Italy 5Earth Observation Science, Department of Physics and Astronomy, University of Leicester, Leicester, UK 6Leicester Institute for Space and Earth Observation, University of Leicester, Leicester, UK 7NERC National Centre for Earth Observation, UK \*Now at: SRON Netherlands Institute for Space Research, Utrecht, the Netherlands

Correspondence: J.D. Maasakkers (maasakkers@fas.harvard.edu)

Abstract. We use 2010-2015 observations of atmospheric methane columns from the GOSAT satellite instrument in a global inverse analysis to improve estimates of methane emissions and their trends over the period, as well as the global concentration of tropospheric OH (the hydroxyl radical, methane's main sink) and its trend. Our inversion solves the Bayesian optimization problem analytically including closed-form characterization of errors. This allows us to (1) quantify the information content

- from the inversion towards optimizing methane emissions and its trends, (2) diagnose error correlations between constraints on 5 emissions and OH concentrations, and (3) generate a large ensemble of solutions testing different assumptions in the inversion. We show how the analytical approach can be used even when prior error standard deviation distributions are log-normal. Inversion results show large overestimates of Chinese coal emissions and Middle East oil/gas emissions in the EDGAR v4.3.2 inventory, but little error in the US where we use a new gridded version of the EPA national greenhouse gas inventory as
- prior estimate. Oil/gas emissions in the EDGAR v4.3.2 inventory show large differences with national totals reported to the 10 United Nations Framework Convention on Climate Change (UNFCCC) and our inversion is generally more consistent with the UNFCCC data. The observed 2010-2015 growth in atmospheric methane is attributed mostly to an increase in emissions from India, China, and areas with large tropical wetlands. The contribution from OH trends is small in comparison. We find that the inversion provides strong independent constraints on global methane emissions (546 Tg  $a^{-1}$ ) and global mean OH
- concentrations (atmospheric methane lifetime against oxidation by tropospheric OH of  $10.8 \pm 0.4$  years), indicating that 15 satellite observations of atmospheric methane could provide a proxy for OH concentrations in the future.

**1 Introduction**

Methane is an important greenhouse gas with a particularly strong decadal climate impact (Stocker et al., 2013). The atmospheric methane concentration has increased by a factor of 2.5 since pre-industrial times (Hartmann et al., 2013). This increase is not well understood but must-is most likely to be mainly driven by anthropogenic sources-activities including the oil/gas in-

- 5 dustry, coal mining, livestock, landfills, wastewater treatment, biomass burning, and rice cultivation (Dlugokencky et al., 2011; Kirschke et al., 2013; Saunois et al., 2016). Wetlands are the main natural source and could be affected by climate change (Kirschke et al., 2013). Atmospheric methane has a lifetime of 9.1 ± 0.9 years (Prather et al., 2012), with a dominant sink from oxidation by the hydroxyl radical (OH) that is also subject to interannual variability and trends (Holmes et al., 2013). Methane concentrations. The methane burden rose by ~ 1-%-12 ppb a-1 in the 1980s and early 1990slate 80s and by ~ 6 
[revised manuscript text omitted]

 (2)

where x is the state vector,  $x_a$  is the prior estimate,  $S_a$  is the prior error covariance matrix, F(x) is the simulation of observations y by the GEOS-Chem model,  $S_O$  is the observational error covariance matrix, and  $\gamma$  is a regularization factor (Brasseur and Jacob, 2017). The variances in  $S_O$  are underestimated because of correlation in the observational error that is missing in the diagonal formulation of  $S_O$  and is difficult to quantify. We use  $\gamma$  to scale the original diagonal  $S_O$  to get an optimal covariance

- 10 matrix to be used in the inversion. Zhang et al. (2018) showed in an observing system simulation experiment (OSSE) for inversion of methane satellite data that a regularization factor  $\gamma = 0.05$  was needed to prevent overfittingbecause of correlation in the observational error that is missing from the diagonal formulation of SO and is otherwise difficult to quantifyadjusts the variances optimally and prevents overfitting. This was done by calculating the likelihood at  $\hat{x}$  for a range of values of  $\gamma$ . Diagnosis of overfit and optimization of  $\gamma$  is readily done in an OSSE such as in Zhang et al. (2018) where the "true" solution is
- 15 known. Here we find that using  $\gamma = 1$  (as in the pure Bayesian statement of the optimization problem) produces checkerboard patterns in the solution that are likely spurious. We choose  $\gamma = 0.05$  consistent with Zhang et al. (2018) for our base inversion as providing the best balance between prior and observational terms in the posterior value of the cost function, and . We examine the sensitivity to the choice of  $\gamma$  by conducting a sensitivity inversion with  $\gamma = 0.1$ .
- Further balancing of the cost function is needed because the global OH concentration and its interannual variability are represented by only 7 state vector elements, while the emissions on the 4° × 5° grid are represented by 1009 elements. To provide equal weight to OH and emissions for explaining global methane trends, we increase the weight of the OH terms in the cost function (through the OH components of Sa) by the ratio of the number of state vector elements 1009/7 so that from a cost-function perspective, a change in OH and global methane emissions are equally expensive. The sensitivity inversion assuming 10% prior error standard deviation on OH instead of 3% is equivalent to decreasing this weighting by a factor of 11.

The GEOS-Chem forward model y = F(x) relating methane column concentrations y to the state vector x is essentially

linear. There is a small non-linearity from the optimization of OH concentrations because changes in the methane concentrations affect the loss rate (Houweling et al., 2017) which we neglect because changes in methane concentrations are small and methane is well-mixed globally. We therefore express the forward model as  $\mathbf{F}(\mathbf{x}) = \mathbf{K}\mathbf{x} + \mathbf{c}$  where  $\mathbf{K} = \partial y/\partial x$  is the Jacobian

30 methane is well-mixed globally. We therefore express the forward model as  $\mathbf{F}(\mathbf{x}) = \mathbf{K}\mathbf{x} + \mathbf{c}$  where  $\mathbf{K} = \partial y / \partial x$  is the Jacobian matrix of the model and  $\mathbf{c}$  is an initialization constant (January 2009 concentrations taken from Turner et al. (2015)). Replacing  $\mathbf{F}(\mathbf{x}) = \mathbf{K}\mathbf{x}$  in Equation 2 and subtracting the initialization constant **c** from the observations, the minimization problem  $dJ(\mathbf{x})/d\mathbf{x} = \mathbf{0}$  has an analytical solution for the optimal posterior solution  $\hat{\mathbf{x}}$  (Rodgers, 2000):

$$\widehat{\mathbf{x}} = \mathbf{x}_{\mathbf{a}} + \mathbf{S}_{\mathbf{a}} \mathbf{K}^T \left( \mathbf{K} \mathbf{S}_{\mathbf{a}} \mathbf{K}^T + \frac{\mathbf{S}_{\mathbf{O}}}{\gamma} \right)^{-1} (\mathbf{y} - \mathbf{K} \mathbf{x}_{\mathbf{a}})$$
(3)

5 The posterior error covariance matrix  $\widehat{\mathbf{S}}$  describing the error statistics of  $\widehat{\mathbf{x}}$  is given by:

$$\widehat{\mathbf{S}} = \left(\gamma \mathbf{K}^T \mathbf{S}_{\mathbf{O}}^{-1} \mathbf{K} + \mathbf{S}_{\mathbf{a}}^{-1}\right)^{-1} \tag{4}$$

and the averaging kernel matrix ( $\mathbf{A} = \partial \hat{\mathbf{x}} / \partial \mathbf{x}$ ) defining the sensitivity of the solution to the true state is given by:

$$\mathbf{A} = \mathbf{S}_{\mathbf{a}} \mathbf{K}^{T} \left( \mathbf{K} \mathbf{S}_{\mathbf{a}} \mathbf{K}^{T} + \frac{\mathbf{S}_{\mathbf{O}}}{\gamma} \right)^{-1}$$
(5)

The trace of the averaging kernel matrix defines the degrees of freedom for signal (DOFS) of the inversion, that is the number 10 of pieces of information on the state vector that can be gained from the observing system.

The analytical solution as described by Equations 3-5 requires the explicit construction of the Jacobian matrix  $\mathbf{K}$  characterizing the GEOS-Chem model. We do this column-by-column with GEOS-Chem simulations perturbing independently each element of the state vector. This is readily achievable even for 2025 state vector elements as a massively parallel computation.

15 Sparse matrix algebra is used where possible in solving Equations 3-5, taking advantage of the diagonal structure of the error covariance matrices.

The analytical solution to the Bayesian optimization problem requires assumption of Gaussian errors, but this allows for the possibility of negative values of state vector elements. Small negative Negative emissions could conceivably be attributed to
soil uptake, but large negative emissions are most likely locally strong soil uptake or oxidation by Cl atoms, but may also be unphysical (Miller et al., 2014). We can address this problem enforce positivity in the Bayesian solution by optimizing for ln(x) instead of x, with normal Gaussian errors specified for ln(x) (corresponding to log-normal errors for x). The model is then non-linear, so that the solution and the corresponding error statistics must be found iteratively with an updated Jacobian matrix K'N = ∂y/∂ln x at each iteration N. This recomputation is immediate using the previously derived Jacobian matrix K for the linear problem, since the individual scalar elements ∂yi/∂ln(xi) of K' are related to those of K by ∂yi/∂ln(xj) = xj∂yi/∂xj. Thus only a simple scaling of the linear Jacobian matrix is required at each iteration. This conversion to log space is done only

- for the emissions component of x. Emission trends and global OH concentrations are still optimized with normal error distributions and no scaling is applied to those rows of the Jacobian.
- 30 Optimizing emissions in log space means that the best posterior estimate is for the median of emissions instead of the mean. The mean and the median of the log-normal distribution are not equal, so that results cannot be summed over grid squares to provide a best estimate of the mean. For this reason, analysis of aggregate and global emissions and sinks will be done with

10

15

The iterative solution for the inverse problem with lognormal errors is obtained with the Levenberg-Marquardt method 5 (Rodgers, 2000) for each iteration N:

$$\mathbf{x}'_{N+1} = \mathbf{x}'_{N} + \left( (1+\kappa) \mathbf{S}'_{\mathbf{A}}^{-1} + \gamma \mathbf{K}'_{N}^{T} \mathbf{S}_{\mathbf{O}}^{-1} \mathbf{K}'_{N}^{-1} \right)^{-1} \left( \gamma \mathbf{K}'_{N}^{T} \mathbf{S}_{\mathbf{O}}^{-1} \left( \mathbf{y} - \mathbf{K} \mathbf{x}_{N} \right) - \mathbf{S}'_{\mathbf{A}}^{-1} \left( \
[revised manuscript text omitted]